# Adaptive Kernel Design for Bayesian Optimization Is a Piece of CAKE with LLMs

**Richard Cornelius Suwandi**[1]    **Feng Yin**[1][*]    **Juntao Wang**[1]
**Renjie Li**[2,3]    **Tsung-Hui Chang**[1]    **Sergios Theodoridis**[4]

[1]School of Artificial Intelligence (SAI), The Chinese University of Hong Kong, Shenzhen
[2]School of Science and Engineering (SSE), The Chinese University of Hong Kong, Shenzhen
[3]Holonyak Micro & Nanotechnology Lab, University of Illinois at Urbana-Champaign
[4]HERON CoE for Robotics and AI-ATHENA R.C., Greece
`{richardsuwandi, juntaowang}@link.cuhk.edu.cn`
`{yinfeng, changtsunghui}@cuhk.edu.cn, renjie2@illinois.edu, stheodor@di.uoa.gr`

## Abstract

The efficiency of Bayesian optimization (BO) relies heavily on the choice of the Gaussian process (GP) kernel, which plays a central role in balancing exploration and exploitation under limited evaluation budgets. Traditional BO methods often rely on fixed or heuristic kernel selection strategies, which can result in slow convergence or suboptimal solutions when the chosen kernel is poorly suited to the underlying objective function. To address this limitation, we propose a freshly-baked Context-Aware Kernel Evolution (CAKE) to enhance BO with large language models (LLMs). Concretely, CAKE leverages LLMs as the crossover and mutation operators to adaptively generate and refine GP kernels based on the observed data throughout the optimization process. To maximize the power of CAKE, we further propose BIC-Acquisition Kernel Ranking (BAKER) to select the most effective kernel through balancing the model fit measured by the Bayesian information criterion (BIC) with the expected improvement at each iteration of BO. Extensive experiments demonstrate that our fresh CAKE-based BO method consistently outperforms established baselines across a range of real-world tasks, including hyperparameter optimization, controller tuning, and photonic chip design. Our code is publicly available at `https://github.com/richardcsuwandi/cake`.

## 1   Introduction

Many important scientific and engineering problems require optimizing objective functions that are noisy and expensive to evaluate. These objective functions often lack closed-form expressions, let alone gradient information, making optimization particularly difficult [50]. Nonetheless, Bayesian optimization (BO) has shown remarkable success in optimizing such functions, due to its ability to operate on limited data and incorporate prior knowledge to guide the optimization process [17]. In the past couple of decades, BO has been used for diverse tasks ranging from tuning hyperparameters in machine learning [43, 29] to designing policies in robotics [9, 33] and recommending new molecules in drug discovery [25, 49]. The main idea behind BO is to first construct a *surrogate model*, typically using a Gaussian process (GP) [36], to represent the prior belief about the objective function. Then, by conditioning on the observations and the prior, the posterior is calculated using Bayes' rule to reflect the updated belief about the objective function. Based on this posterior, an *acquisition function* is further used to determine the next promising query positions while balancing exploration (i.e., moving to regions with high uncertainty) and exploitation (i.e., moving to regions with high expected value).

---

[*]Corresponding author.

39th Conference on Neural Information Processing Systems (NeurIPS 2025).

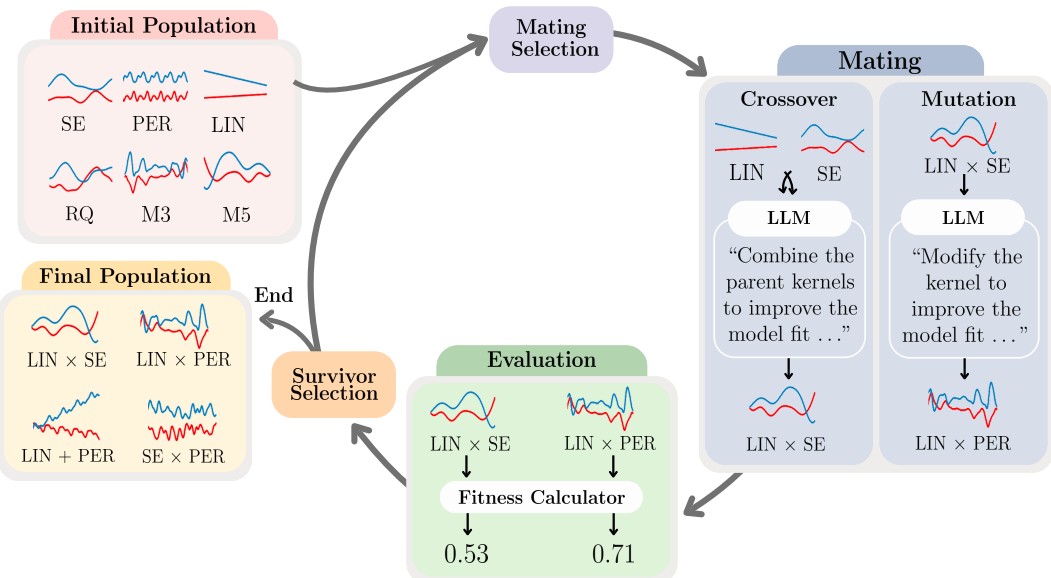

Figure 1: Overview of CAKE. Starting with an initial population of kernels, the LLM acts as crossover and mutation operators, proposing new kernels based on the given prompts. The proposed kernels are then evaluated using a fitness calculator, and the fittest ones advance to the next generation.

Although the past decades have witnessed rapid development of BO, much of the focus has been drawn on designing novel acquisition functions [2, 1]. In contrast, the challenge of appropriately choosing the surrogate model has received comparatively less attention [40]. In the context of GPs, most off-the-shelf BO methods simply use general-purpose kernels, such as the squared exponential kernel or Matérn-5/2 kernel [43]. While convenient, this *one-size-fits-all* approach may introduce bias that can negatively impact the sampling of potential solutions during optimization [15], especially when the kernel's assumptions do not align with the statistical properties of the underlying objective function [37]. It has also been studied that with a poor choice of the kernel, BO may converge very slowly, especially when optimizing complex functions in moderate-to-high dimensional spaces [16]. These considerations underscore the need for a more sophisticated kernel design in BO.

While the pursuit for automatic kernel design is not new [12, 54], existing approaches might not be straightforward in the setting of BO. In BO, the objective function is typically expensive to evaluate, limiting the number of evaluations we can perform and leaving us with a small number of observations to tune the kernel. Moreover, since the functional form is generally unknown and the gradient information is unavailable, it becomes infeasible to apply traditional optimization-based kernel selection methods, such as [4, 19]. At its core, these challenges can be framed within the *few-shot learning* setting, where swift learning and generalization from limited data are required. Notably, these challenges align with the strengths of large language models (LLMs) [52], which excel at generalizing from few-shot samples [8], thus enabling efficient exploration with limited data. The *in-context learning* capability of LLMs also acts as implicit Bayesian inference [56, 20], allowing them to encode prior knowledge about the optimization task, search space, and other relevant information. LLMs' ability in performing complex reasoning further enhances their capacity to process contextual information and improve search performance [57]. Moreover, LLMs are also pre-trained on massive internet data, which potentially contains transferable domain knowledge applicable to various optimization tasks [29]. Given these insights, we aim to investigate the following question: "*Can LLMs, with their encoded knowledge and few-shot prompting, help to adaptively evolve kernel structures based on the observed data, thereby improving the BO performance?*"

**Contributions.** In this paper, we introduce Context-Aware Kernel Evolution (CAKE) to enable adaptive BO using LLMs. Unlike conventional BO setups, which often rely on fixed kernels or heuristic selection strategies, CAKE leverages the in-context learning capabilities of LLMs to iteratively generate and refine expressive kernel structures based on the acquired data during the

optimization process (see Figure 1). CAKE is guided by few-shot prompting and operates entirely in-context, without requiring fine-tuning or parameter updates to the LLM. To further complement CAKE, we propose BIC-Acquisition Kernel Ranking (BAKER) to select the most effective kernel through balancing the model fit measured by the Bayesian information criterion (BIC) with the expected improvement at each iteration of BO. We demonstrate through extensive experiments that our proposed method consistently outperforms established baselines across a range of real-world optimization tasks, including optimizing the hyperparameters of machine learning models, tuning controllers for dynamic environments, and designing photonic chips with optimal configurations.

## 2 Preliminaries

We first review some key concepts in BO and the so-called *kernel grammar*, which serve as the bedrock for this paper.

**Bayesian optimization.** Consider optimizing a "black-box" objective function $f : \mathcal{X} \subset \mathbb{R}^d \to \mathbb{R}$, where the function evaluations are noisy, expensive, and the gradients are not available. Bayesian optimization (BO) addresses these challenges by employing a probabilistic surrogate model $g$, typically using a Gaussian processes (GP) [36], to approximate $f$ on the fly [17]. At each iteration $t$, the calibration of the posterior distribution $p(g_t \mid \mathcal{D}_t)$ given the previous observations $\mathcal{D}_t = \{(\mathbf{x}_i, y_i)\}_{i=1}^{t-1}$ informs where to explore and exploit in $\mathcal{X}$. Then, based on $p(g_t \mid \mathcal{D}_t)$, an acquisition function $\alpha : \mathcal{X} \to \mathbb{R}$ defines a policy to choose the next best point to evaluate. After each evaluation, the surrogate model $g$ is refined to reflect the updated belief about $f$. We refer the readers to Appendix A.1 for a thorough treatment of BO with GPs.

**Kernel grammar.** The kernel grammar introduced by Duvenaud et al. [12] defines a comprehensive and flexible space of kernels. Such space exploits the closure properties of kernel functions under addition and multiplication, which ensures that: if $k_1(\mathbf{x}, \mathbf{x}')$ and $k_2(\mathbf{x}, \mathbf{x}')$ are valid kernels, then both $k_1(\mathbf{x}, \mathbf{x}') + k_2(\mathbf{x}, \mathbf{x}')$ and $k_1(\mathbf{x}, \mathbf{x}') \times k_2(\mathbf{x}, \mathbf{x}')$ are also valid kernels [42]. Starting from a set of base kernels, such as the squared exponential (SE), linear (LIN), and periodic (PER) kernels, one can construct more expressive kernels by combining such base kernels via addition and multiplication operators. For instance, LIN + PER kernel can capture periodic structure with a linear trend and SE $\times$ PER kernel can capture locally periodic components. Let $\mathcal{B}$ denote a base kernel and $\mathcal{S}$ denote a subexpression. For example, in the expression LIN + (PER $\times$ SE), the term PER $\times$ SE is a subexpression. The entire kernel space is described by all kernels that can be constructed by adding a base kernel to a subexpression $\mathcal{S} \to \mathcal{S} + \mathcal{B}$, multiplying a subexpression with a base kernel $\mathcal{S} \to \mathcal{S} \times \mathcal{B}$, and replacing a base kernel with another base kernel $\mathcal{B} \to \mathcal{B}$.

## 3 Context-Aware Kernel Evolution

The main motivation behind our method is to refine the kernel choice at each iteration before determining the next query point. To this end, we introduce Context-Aware Kernel Evolution (CAKE), which leverages LLMs as genetic operators to adaptively construct kernels based on the data obtained on the fly. We summarize the complete procedure of CAKE in Algorithm 1.

**Conditioning the LLM.** We begin by randomly sampling $n$ points from the input space $\mathcal{X}$ to initialize the observations $\mathcal{D} = \{(\mathbf{x}_i, y_i)\}_{i=1}^n$. These observations are then used as *few-shot samples* to prompt the LLM. The prompt is designed based on the concept of conditioning on high performance, as suggested by [60]. Specifically, we start the prompt with a statement, *"You are an expert in machine learning, specializing in Gaussian processes",* to simulate the reasoning of a human expert in the field. It has also been shown that *chain-of-thought reasoning*, or generating intermediate reasoning steps, can improve the performance of LLMs [53, 24]. Motivated by this, we instruct the LLM to analyze the provided observations and identify patterns that can be represented by kernel functions, before proposing the kernels at each iteration. The designed system prompt is shown in Figure 2.

**Initializing the population.** We draw some inspiration from the genetic algorithm [22], where we maintain a population of candidates (kernels) throughout the optimization process. We define $\mathbb{K}$ as our population, and for each kernel $k \in \mathbb{K}$, we measure its *fitness* using the Bayesian Information Criterion (BIC) [39]. The BIC is a widely used metric for model selection that measures the trade-off between model fit and model complexity [46]. It can also be viewed as an approximation of the Laplace method for estimating the marginal likelihood [35]. We provide more detailed discussions

Figure 2: The designed system prompt. `{}` indicate placeholders.

(a) Crossover prompt.                          (b) Mutation prompt.

Figure 3: Prompts for evolving the kernels via crossover and mutation.

on BIC and model selection for GPs in Appendix A.2. To ensure the fitness score is consistent across different tasks, we normalize it to the range $[0, 1]$.

**Proposing the kernels.** We consider a generalized notion of the kernel grammar [7], which involves a set of base kernels $\{k_1, \ldots, k_r\}$ and a set of operators $\{\mathcal{T}_1, \ldots, \mathcal{T}_l\}$, where $r, l \in \mathbb{N}$. Each operator $\mathcal{T}_j : \mathcal{K} \times \mathcal{K} \to \mathcal{K}$, for $j = 1, \ldots, l$, is a closed operator (e.g., addition, multiplication, convolution, composition, affine transformation) on the space of kernels $\mathcal{K}$. Based on this, we can define the kernel grammar space recursively as follows: $\mathbb{K}_0 := \{k_1, \ldots, k_r\}$ and $\mathbb{K}_i := \{\mathcal{T}_j(k_1, k_2) \mid k_1, k_2 \in \mathbb{K}_{i-1}, j = 1, \ldots, l\} \cup \mathbb{K}_{i-1}$, for $i \in \mathbb{N}$. By leveraging this kernel grammar space, we can leverage the LLM as *genetic operators* to propose kernels using the following operations:

1. **Crossover:** We perform $n_c$ crossover operations. For each crossover, we sample two parent kernels $k_1, k_2$ from $\mathbb{K}$ with probability proportional to their fitness. We then prompt the LLM to propose a new kernel $k_c$ by applying an operator on the parent kernels.

2. **Mutation:** With probability $p_m$, we perform a mutation operation. We select the fittest kernel $k_f$ from $\mathbb{K}$ and prompt the LLM to suggest a new kernel $k_m$, by replacing one of the base kernels in $k_f$ with another base kernel.

The sample prompts for the crossover and mutation operations are shown in Figure 3. In the prompts, we also ask the LLM to report its reasoning behind the proposed kernels (see Appendix C.3 for a sample response). This serves as a sanity check, enabling us to verify and interpret the choices made by the LLM. The proposed kernels from the two operations are added to $\mathbb{K}$, and their fitnesses are measured. Then, we select the top $n_p$ fittest kernels to form the next generation of $\mathbb{K}$.

**Choosing the next query point.** In our experiments, we observed that some kernels may promise a good fit, but the actual improvement from the query points they propose is not as substantial as expected. For this reason, we propose the BIC-Acquisition Kernel Ranking (BAKER) to jointly rank kernels based on both their model fit and their potential to yield high-utility query points. We first assign weights to each kernel $k \in \mathbb{K}$ based on its BIC: $w_k = \exp(-\mathrm{BIC}_k)/\sum_{k' \in \mathbb{K}} \exp(-\mathrm{BIC}_{k'})$, where $\mathrm{BIC}_k$ is the BIC value of the GP model associated with kernel $k$. We denote the acquisition

**Algorithm 1** Context-Aware Kernel Evolution (CAKE)

---

**Require:** Budget $T$, number of crossovers $n_c$, mutation probability $p_m$, population size $n_p$
 1: Randomly sample $n$ points to form the initial observations $\mathcal{D} = \{(\mathbf{x}_i, y_i)\}_{i=1}^n$
 2: Initialize the kernel population $\mathbb{K}$ with the set of base kernels
 3: **for** $t = 1$ to $T$ **do**
 4:     Update system prompt with $\mathcal{D}$ (see Fig. 2)
 5:     **for** $c = 1$ to $n_c$ **do**
 6:         Sample two parent kernels $k_1, k_2$ from $\mathbb{K}$
 7:         Generate new kernel $k_c$ via crossover (see Fig. 3a)
 8:     **end for**
 9:     **if** `rand()` $< p_m$ **then**
10:         Select the fittest kernel in $\mathbb{K}$
11:         Generate new kernel $k_m$ via mutation (see Fig. 3b)
12:     **end if**
13:     Evaluate the fitnesses and keep the top-$n_p$ kernels in $\mathbb{K}$
14:     Choose the most effective kernel $k^*$ via BAKER (see Eq. 1)
15:     Obtain the next point $\mathbf{x}_t = \mathbf{x}_{t,k^*}$ and evaluate $y_t = f(\mathbf{x_t})$
16:     Update the observations as $\mathcal{D} \leftarrow \mathcal{D} \cup \{(\mathbf{x}_t, y_t)\}$
17: **end for**

---

function[2] as $\alpha(\mathbf{x}; \mathcal{D}, k)$, which quantifies the utility of evaluating a candidate point $\mathbf{x}$ under the model that kernel $k$ is being used, given the current observations $\mathcal{D}$. Based on this, BAKER computes a weighted acquisition value for each kernel and selects the kernel $k^*$ that maximizes this value, i.e.,

$$k^* = \arg\max_{k \in \mathbb{K}} w_k \alpha(\mathbf{x}_{t,k}; \mathcal{D}, k), \tag{1}$$

where $\mathbf{x}_{t,k}$ denotes the candidate query point proposed by kernel $k$ at iteration $t$. BAKER allows us to balance the kernel's ability to fit the data (as indicated by $w_k$) with the expected improvement at the proposed query point (as measured by $\alpha$). Once $k^*$ is selected, we use the corresponding kernel to obtain the next query point $\mathbf{x}_t = \mathbf{x}_{t,k^*}$, evaluate $y_t = f(\mathbf{x}_t)$, and update the observations as $\mathcal{D} \leftarrow \mathcal{D} \cup \{(\mathbf{x}_t, y_t)\}$. This iterative process continues until a predefined budget $T$ is exhausted.

## 4 Related Work

**Expressive kernel design.** Several methods have been developed to construct more expressive kernels beyond manual composition of base kernels. One such method involved multiple kernel learning techniques [4, 19], which aim to identify the optimal kernel configuration by optimizing a linear or nonlinear combination of base kernels. However, these methods restrict the kernel space and require prior specification of the kernel hyperparameters. Another approach involved searching for the optimal kernel structure across a space of kernels [12], but since the space is infinite, efficiently navigating this space demands modeling expertise. Other works focus on designing flexible kernel families via spectral approximations [26, 54], or integrating GPs with deep neural networks [55]. While powerful, these approaches either assume stationarity or require complex inference techniques. In contrast, our method is based on the kernel grammar and in-context learning via LLMs, which offers a flexible yet computationally feasible approach.

**Surrogate modeling in BO.** When using GPs as the surrogate model in BO, the kernel is typically selected *a priori* based on an expert's knowledge concerning the problem at hand. Unfortunately, if there is no prior knowledge available, most BO methods simply use default kernels such as the SE kernel or the Matérn-5/2 kernel [43]. While this seems reasonable, it has been reported that with poor or overly general choices of the kernel, BO may converge very slowly [16]. For this reason, deep GPs have been proposed to help model non-stationary behaviors [21], but at the cost of increased computational complexity. Other works explore adaptive kernel strategies, such as using discrete mixtures of GPs [18], maintaining parallel GPs with different kernels [37], or using ensembles of GP [31]. Recent works have also showed great potential in using LLMs for surrogate modeling in

---

[2]We use the expected improvement (EI) as our default acquisition function, normalized to $[0, 1]$ to ensure comparability across different kernels.

BO [29, 11, 59]. We extend this line of research by using LLMs to automatically generate and refine kernels during the optimization process, enabling a new-fashioned adaptive kernel design.

**LLMs as genetic operators.** As the model size and amount of training data increase, LLMs exhibit emergent abilities that significantly improve their performance across diverse tasks [52, 8]. Inspired by these abilities, recent works have explored using LLMs as genetic operators for generating code [34], assisting robot simulations [27], and designing neural network architectures [10]. To the best of our knowledge, the current work is the first to use LLMs as genetic operators for constructing adaptive and expressive GP kernel design for BO. Compared to the other transformer-based methods [41], our method can be applied entirely in-context and does not require any fine-tuning.

## 5 Experiments

To evaluate the performance of our proposed method, we test it against several baselines across a set of real-world optimization tasks with varying characteristics, including diverse optimization landscapes, dynamic environments, and multi-objective settings.

**Setup.** Our experiments were conducted using the software package BoTorch [6] and we used the expected improvement (EI) as our default acquisition function. For the LLM, we use OpenAI's `gpt-4o-mini` model as it offers an excellent balance between API cost affordability, fast inference speed, and intelligence for our implementation. We define $\{\text{SE}, \text{PER}, \text{LIN}, \text{RQ}, \text{M3}, \text{M5}\}$ as our base kernels and $\{+, \times\}$ as our operators. Moreover, we set the number of crossovers $n_c = 5$, mutation probability $p_m = 0.7$, and population size $n_p = 10$. To facilitate reproducibility, our code is available online at `https://github.com/richardcsuwandi/cake`. The shaded regions in all figures represent the standard error over independent trials. In the interest of space, more experimental details can be found in Appendix B, and additional results are provided in Appendix C.

**Baselines.** We compare our proposed method against the following established baselines:

- **Fixed:** Default method in BO, where we fix the kernel throughout the optimization process.
- **Adaptive:** An adaptive kernel selection method proposed in [37]. We employ three different selection criteria to adaptively change the kernel: *Random*, *Utility*, and *BIC*.
- **Deep GP:** Uses a deep GP (DGP) as the surrogate model, implemented through a functional composition of stationary GPs [21].
- **Ensemble GP:** Uses an ensemble of GPs (EGP) to adaptively select the surrogate model [31]. The kernel dictionary consists of the same six kernels used in CAKE.
- **Compositional Kernel Search (CKS):** Uses greedy search to discover kernel structures that best explains the observed data [12].
- **Automated BO (ABO):** Treats the kernel selection as a "black-box" optimization problem and uses BO to solve it [32].

### 5.1 Hyperparameter Optimization

**Setup.** We consider the hyperparameter optimization tasks available in the HPOBench package [14]. We included a total of 60 tasks, comprising 12 OpenML datasets and 5 machine learning (ML) models: logistic regression (LR), support vector machine (SVM), random forest (RF), XGBoost (XGB), and multi-layer perceptron (MLP). Each model-dataset pair exhibits a unique optimization landscape, making these tasks an ideal testbed for evaluating the generalization performance of BO methods across diverse scenarios. Further details regarding the hyperparameter search spaces and dataset characteristics are provided in Appendix B.3.1. Here, our goal for each task is to maximize the accuracy of the ML model on the unseen test data. Each experiment was executed for $T = 100$ trials and repeated using 20 different random seeds.

**Results.** Figure 4 shows the average test accuracy for different ML models on all datasets. The results demonstrate that CAKE consistently achieves the highest accuracy compared to the other methods across all tasks. It is also worth noting that CAKE excels in the earlier stages of the optimization process, when fewer observations are available. This suggests that CAKE is able to effectively leverage fewer data samples to quickly converge to high-performing configurations. We provide a quantitative analysis to support this finding in Appendix C.7. Our results also reveal significant

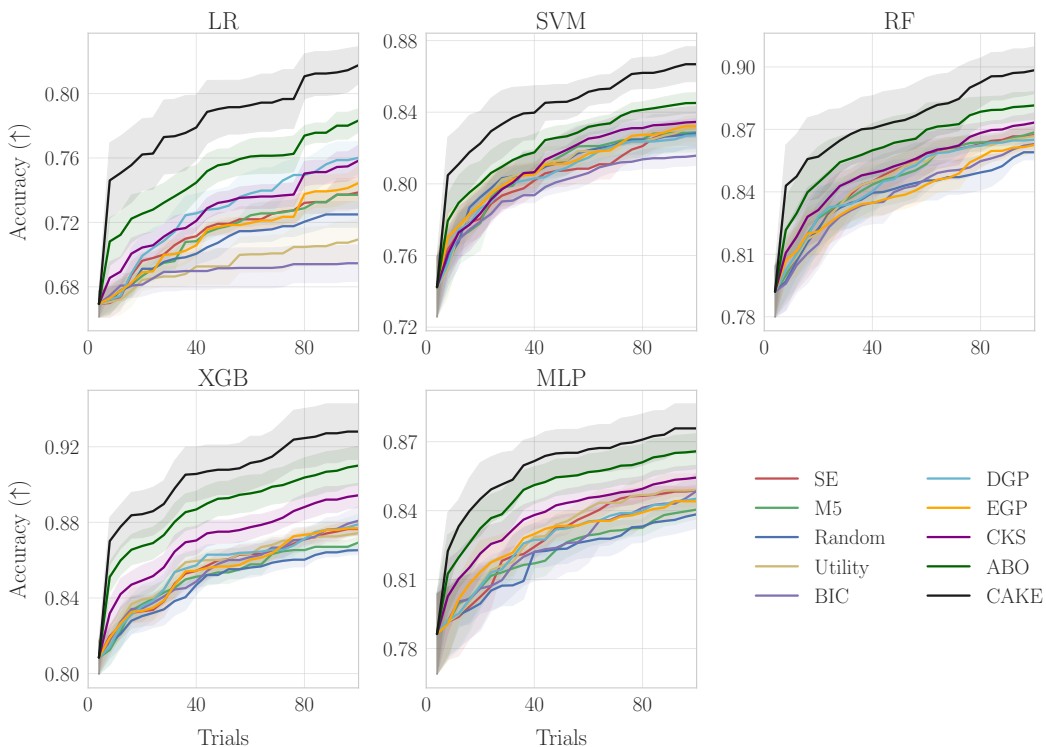

Figure 4: Average test accuracy over 20 random seeds for different ML models.

variations in performance among fixed and adaptive kernel methods. For instance, M5 and Utility perform reasonably well in tuning SVM and RF models, but struggle with tuning LR and XGB. EGP and DGP exhibit moderate performance, often outperforming fixed kernels but falling short compared to more flexible approaches such as CKS and ABO. Overall, CAKE demonstrates superior performance consistently across all tasks. Another key advantage of CAKE is that the learned kernel expressions are also interpretable. In Appendix C.4, we analyze one such expression and show how CAKE automatically translates it into a natural language description.

## 5.2 Controller Tuning

**Setup.** We consider two real-world controller tuning tasks that simulate dynamic environments, where small changes in the environment condition may result in significantly different outcomes. For the first task, we consider the robot pushing problem [51], which involves tuning a controller for two robotic hands to push two objects towards some specified target positions. The controller is parameterized by $d = 14$ parameters that determine the position and orientation of the hands, the pushing speed, direction of movement, and duration of the push. The second task involves tuning a controller for the lunar lander environment [48], which is defined by $d = 12$ parameters that determine how to map the 8-dimensional state vector comprising of position, angle, velocity, and ground contact indicators, to one of four actions: firing the main engine, left or right orientation engines, or doing nothing. The goal is to achieve a cumulative reward of at least 200 points, which corresponds to a successful landing while minimizing penalties from crashes or excessive engine use. For both tasks, we evaluate the performance using $T = 1000$ iterations, averaging the results over 10 different initial conditions (e.g., positions, terrains, velocities). We provide more details regarding the reward functions and environment implementations in Section B.3.2.

**Results.** The results shown in Figure 5 demonstrate that the controllers optimized by CAKE achieved the highest average rewards in both tasks. In robot pushing task, CAKE converges to a high-performing solution much faster than the other methods, highlighting its ability to quickly learn effective control policies from limited data. On the other hand, we found that fixed kernels like SE and M5 tend to plateau earlier, while adaptive methods such as Utility and BIC show only moderate

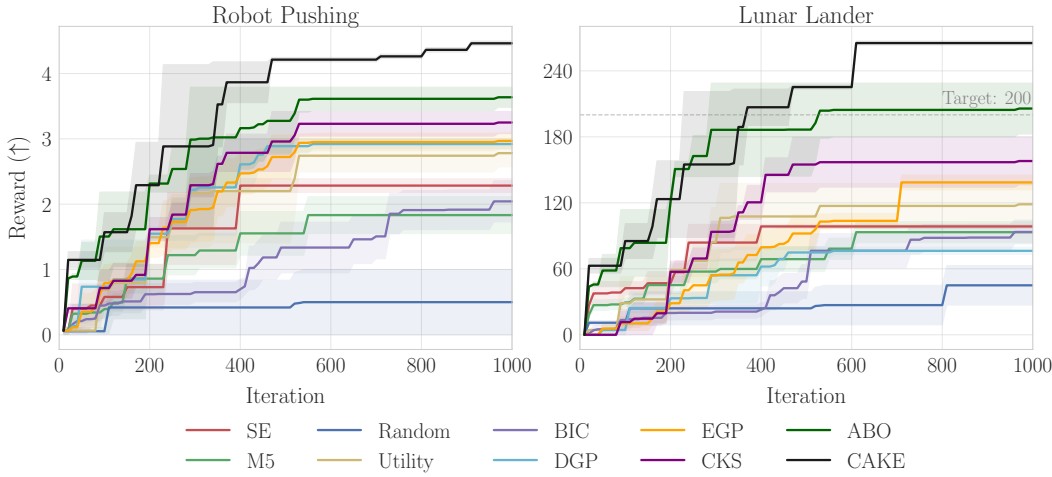

Figure 5: Average reward for the controller tuning tasks over 10 different initial conditions.

improvements over the fixed kernels and still lag behind compositional approaches. Notably, ABO is the only method besides CAKE that is able to reach the target score of 200 points in the lunar landing task. Despite this, we observed that ABO exhibits greater performance fluctuations compared to CAKE, especially in more challenging environments. Meanwhile, the remaining methods struggle to maintain high scores and often incur greater penalties due to crashes or inefficient landings. Overall, these results show that CAKE's adaptive nature enables it to effectively respond to changes in the underlying objective (e.g., environmental shifts), offering greater robustness than fixed-kernel methods that may fail under such conditions.

## 5.3 Photonic Chip Design

**Setup.** We apply our method to the design of photonic chips, which is a challenging problem in physics and engineering [28]. Due to the high cost of fabrication, it is infeasible to try all, or even a few, of the design parameters. As a result, one has to rely on extensive computer simulations to assess the chip's performance. This makes the task a "black-box" inverse design problem, where the goal is to optimize the chip parameters to meet some desired performance indicators. We consider five key indicators for assessing the chip's performance: Q-factor ($f_1$), wavelength ($f_2$), lasing area ($f_3$), power ($f_4$), and divergence angle ($f_5$). Based on these performance indicators, we can calculate the overall score for a given set of parameters: $\alpha f_1 + \beta f_2 + \gamma f_3 + \delta f_4 + \epsilon f_5$, where we set $\alpha = \beta = 1$, $\gamma = \delta = 100$, and $\epsilon = 20$ to unify the scale between different objectives. Our goal is to find a Pareto-optimal solution that balances the trade-offs among the five competing objectives, thereby achieving the best overall chip performance. We provide detailed descriptions of each objective and their physical interpretations in Appendix B.3.3.

**Baselines.** We consider two widely-used multi-objective BO methods: *Single-Task GP*, where each objective is modeled separately using a GP with an M5 kernel, and *Additive GP*, which models the overall objective as a sum of independent GPs, each using an SE kernel. We also include CKS and ABO as compositional kernel baselines to provide a direct comparison against our proposed CAKE method. For all methods, we set $T = 250$ with 10 different random initializations and employed the expected hypervolume improvement (EHVI) [58] as the acquisition function.

**Results.** Figure 6 shows the score and hypervolume of the chip designed by CAKE and the competing baselines. Compared to the baselines, CAKE achieved the highest values in both metrics, indicating superior optimization performance and better exploration of the Pareto front. This likely stems from its ability to utilize different kernels tailored to individual objectives, unlike the other baselines that rely on a single or additive kernel shared across all objectives. CAKE also outperforms CKS and ABO, demonstrating its advantage in handling multi-objective settings. Notably, CAKE found a solution with a significantly higher score in less than 40 trials, which is equivalent to a tenfold speedup in the design cycle compared to the baselines. From a practical point-of-view, this level of acceleration translates to significant reductions in both the time and cost needed to design the chips.

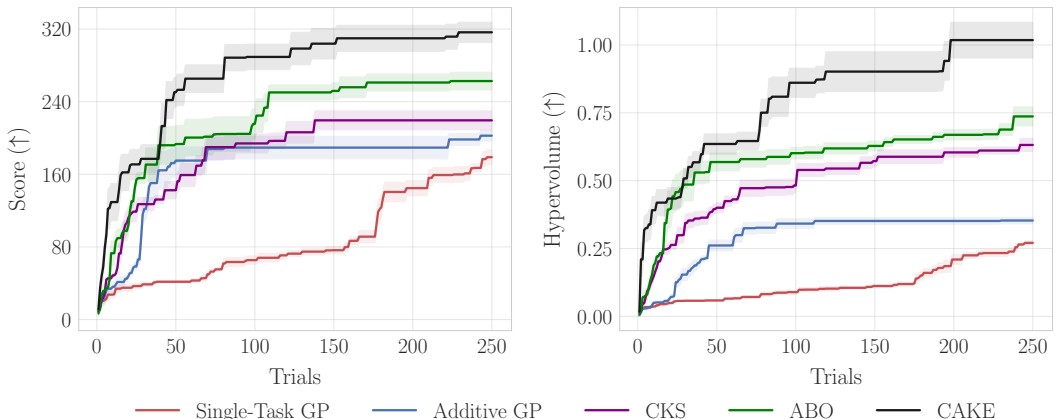

Figure 6: Average score and hypervolume of the designed chip over 250 trials.

Table 1: Average rank ($\downarrow$) $\pm$ standard error on HPOBench over 20 random seeds.

| Method | LR | SVM | RF | XGB | MLP | Average |
|---|---|---|---|---|---|---|
| Random Sampling | $6.8 \pm 0.1$ | $6.9 \pm 0.1$ | $6.7 \pm 0.1$ | $6.8 \pm 0.1$ | $6.8 \pm 0.1$ | 6.80 |
| Genetic Algorithm | $2.6 \pm 0.1$ | $2.5 \pm 0.1$ | $2.7 \pm 0.1$ | $2.8 \pm 0.1$ | $2.9 \pm 0.1$ | 2.70 |
| CAKE + BIC | $3.0 \pm 0.1$ | $3.1 \pm 0.1$ | $2.9 \pm 0.1$ | $3.0 \pm 0.1$ | $3.1 \pm 0.1$ | 3.02 |
| CAKE + Utility | $2.3 \pm 0.1$ | $2.2 \pm 0.1$ | $2.4 \pm 0.1$ | $2.5 \pm 0.1$ | $2.6 \pm 0.1$ | 2.40 |
| Adaptive + BAKER | $4.5 \pm 0.1$ | $4.4 \pm 0.1$ | $4.6 \pm 0.1$ | $4.8 \pm 0.1$ | $4.7 \pm 0.1$ | 4.60 |
| CKS + BAKER | $3.1 \pm 0.1$ | $3.2 \pm 0.1$ | $3.0 \pm 0.1$ | $3.1 \pm 0.1$ | $3.2 \pm 0.1$ | 3.12 |
| CAKE + BAKER | $1.1 \pm 0.1$ | $1.0 \pm 0.1$ | $1.0 \pm 0.1$ | $1.1 \pm 0.1$ | $1.1 \pm 0.1$ | **1.04** |

### 5.4 Ablation Study

**Setup.** To evaluate the influence of each component of our method, we conduct an ablation study with the following configurations: **(1) Random Sampling:** randomly combines base kernels using addition and multiplication, **(2) Genetic Algorithm**: applies genetic operators on a population of kernels guided by fitness, **(3) CAKE + BIC / CAKE + Utility:** CAKE selects kernels based on the fitness or acquisition value only, **(4) Adaptive + BAKER / CKS + BAKER:** adaptive or compositional baselines with BAKER, and **(5) CAKE + BAKER:** full method with LLM and BAKER. We evaluate all ablation setups on the full set of 60 tasks in HPOBench.

**Results.** Table 1 shows that CAKE + BAKER achieves the best overall performance, indicating that both the LLM and BAKER contribute to the effectiveness of our method. We found that removing either component leads to performance degradation. For instance, CAKE + BIC and CAKE + Utility, which only use one selection criterion, perform reasonably well but underperform compared to CAKE + BAKER. This suggests that while the fitness-based or utility-based kernel selection is individually effective, combining them with BAKER improves the performance across diverse tasks. As expected, random sampling performs the worst, suggesting that the LLM generates meaningful kernel expressions rather than just random combinations. We further support this by analyzing the evolution of the kernel population's fitness (see Section 5.5), where we observe a distribution shift toward higher fitness values after each successive round of LLM edits. Although CKS + BAKER outperforms Adaptive + BAKER, it still underperforms compared to CAKE + BAKER. This indicates that while compositional kernels can capture more complex patterns than standard kernels, they still lack the contextual understanding provided by the LLM.

### 5.5 Evolution of Population Fitness

To demonstrate the benefits of using LLM for kernel generation, we conduct an experiment analyzing how the fitness of the kernel population evolves through successive edits by the LLM[3]. Starting with

---

[3]In this context, an LLM edit refers to performing both crossover and mutation operations.

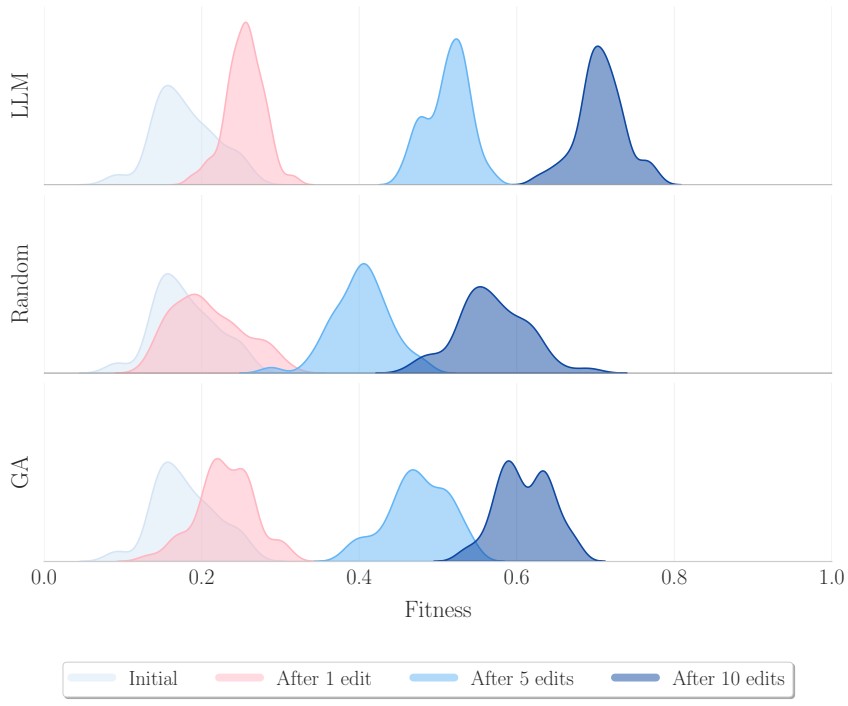

Figure 7: Evolution of the kernel population's fitness over successive edits by LLM, random recombination, and genetic algorithm (GA).

an initial population of base kernels, we evaluate their fitness scores and then iteratively apply LLM edits to evolve the population over time. We further compare it with two baselines which replaces the edit step with random recombination and genetic algorithm (GA) operators. Figure 7 illustrates the progression of the fitness distributions after 1, 5, and 10 successive edits. The initial population exhibits a wide distribution of fitness values with a concentration on lower scores, reflecting the variability in the fitness scores among the base kernels. After just one LLM edit, the distribution immediately shifts toward higher fitness values, indicating that the LLM effectively enhances the population through both crossover and mutation. As more LLM edits are applied, the mean fitness continues to increase, and the variance narrows, indicating consistent improvement and convergence toward high-performing kernels. In contrast, Random and GA show slower convergence, with their fitness distributions remaining broader and less sharply peaked over the same number of edits. in driving population fitness upward. Overall, these results demonstrate that not only LLMs can serve as effective genetic operators, but also outperform those produced by random recombination and traditional GA operators.

## 6 Conclusion

We introduced CAKE, a novel kernel design method that leverages LLMs as crossover and mutation operators to enable adaptive surrogate modeling in BO. To complement CAKE, we further propose BAKER to address the issue where kernels with good model fit may not lead to substantial improvement towards the optimization progress. Experimental results have shown that CAKE achieved superior accuracy across all of the hyperparameter optimization tasks, particularly excelling in the early stages when the observations are scarce. In the dynamic controller tuning tasks, CAKE consistently obtained the highest average rewards and showed strong adaptability under varying environmental conditions. In the photonic chip design problem, CAKE demonstrated its ability to effectively balance multiple conflicting objectives while achieving significant speedup over baseline methods. While the current work focuses on BO, our broader goal is to develop a universal adaptive kernel method that is applicable to various ML tasks. We refer the readers to Appendix D for further discussions on the limitations and directions for future work.

## Acknowledgements

This work was supported in part by the NSFC under Grant 62271433, in part by Shenzhen Science and Technology Program under Grant RCJC20210609104448114 and ZDSYS20230626091302006, and in part by the Guangdong Provincial Key Laboratory of Big Data Computing, The Chinese University of Hong Kong, Shenzhen.

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

# A  Mathematical Details

## A.1  Bayesian Optimization with Gaussian Processes

A Gaussian process (GP) describes a collection of random variables, any finite number of which have a joint Gaussian distribution [36]. Mathematically, a GP can be expressed as, $\mathcal{GP}\left(m(\mathbf{x}), k_{\boldsymbol{\theta}}(\mathbf{x}, \mathbf{x}'; \boldsymbol{\theta})\right)$, where $m(\mathbf{x})$ is the mean function and $k_{\boldsymbol{\theta}}(\mathbf{x}, \mathbf{x}'; \boldsymbol{\theta})$ is the covariance (kernel) function parameterized by some hyperparameters $\boldsymbol{\theta}$. Given any finite collection of inputs $\mathbf{X} = \mathbf{x}_{1:t}$, the outputs are jointly Gaussian,

$$f(\mathbf{X}) \sim \mathcal{N}\left(\boldsymbol{m}_{\mathbf{X}}, \mathbf{K}(\mathbf{X}, \mathbf{X}; \boldsymbol{\theta})\right), \tag{2}$$

where $\boldsymbol{m}_{\mathbf{X}} = m(\mathbf{X}) \in \mathbb{R}^t$ is the mean function vector evaluated at $\mathbf{X}$, often assumed to be $\mathbf{0}$ in practice, and $\mathbf{K}(\mathbf{X}, \mathbf{X}; \boldsymbol{\theta}) \in \mathbb{R}^{t \times t}$ is the covariance matrix with entries $[\mathbf{K}(\mathbf{X}, \mathbf{X}; \boldsymbol{\theta})]_{i,j} = k_{\boldsymbol{\theta}}(\mathbf{x}_i, \mathbf{x}_j)$. We assume that the evaluations of $f$ at any point $\mathbf{x}_t$ are corrupted by a $\sigma$-sub-Gaussian noise [5],

$$y_t = f(\mathbf{x}_t) + \epsilon_t, \tag{3}$$

where $\epsilon_t \sim \mathcal{N}(0, \sigma_\epsilon^2)$. Given the observed data $\mathcal{D}_t = \{\mathbf{X}, \mathbf{y}\}$, where $\mathbf{y} = \mathbf{y}_{1:t}$, the joint Gaussian distribution of the observed data and an arbitrary query point $\mathbf{x}$ is:

$$\begin{bmatrix} \mathbf{y} \\ f(\mathbf{x}) \end{bmatrix} \sim \mathcal{N}\left(\mathbf{0}, \begin{bmatrix} \mathbf{K}_{t;\boldsymbol{\theta}} + \sigma_\epsilon^2 \mathbf{I} & \mathbf{k}_{t;\boldsymbol{\theta}}(\mathbf{x}) \\ \mathbf{k}_{t;\boldsymbol{\theta}}^\top(\mathbf{x}) & k_{\boldsymbol{\theta}}(\mathbf{x}, \mathbf{x}) \end{bmatrix}\right), \tag{4}$$

where $\mathbf{K}_{t;\boldsymbol{\theta}} = \mathbf{K}(\mathbf{X}, \mathbf{X}; \boldsymbol{\theta})$ and $\mathbf{k}_{t;\boldsymbol{\theta}}(\mathbf{x}) = \mathbf{k}_{\boldsymbol{\theta}}(\mathbf{X}, \mathbf{x})$. It follows that, the posterior distribution of any query point $\mathbf{x}$ is marginally Gaussian,

$$f(\mathbf{x})|\mathcal{D}_t; \boldsymbol{\theta} \sim \mathcal{N}(\mu_t(\mathbf{x}; \boldsymbol{\theta}), \sigma_t^2(\mathbf{x}; \boldsymbol{\theta})), \tag{5}$$

where

$$\mu_t(\mathbf{x}; \boldsymbol{\theta}) = \mathbb{E}[f(\mathbf{x})|\mathcal{D}_t] = \mathbf{k}_{t,\boldsymbol{\theta}}^\top(\mathbf{x})(\mathbf{K}_{t;\boldsymbol{\theta}} + \sigma_\epsilon^2 \mathbf{I})^{-1}\mathbf{y}, \tag{6a}$$

$$\sigma_t^2(\mathbf{x}; \boldsymbol{\theta}) = \mathbb{E}[f(\mathbf{x})f(\mathbf{x})|\mathcal{D}_t] = \mathbf{k}_{\boldsymbol{\theta}}(\mathbf{x}, \mathbf{x}) - \mathbf{k}_{t,\boldsymbol{\theta}}^\top(\mathbf{x})(\mathbf{K}_{t;\boldsymbol{\theta}} + \sigma_\epsilon^2 \mathbf{I})^{-1}\mathbf{k}_{t,\boldsymbol{\theta}}(\mathbf{x}). \tag{6b}$$

Based on the above posterior distribution, the acquisition function the use its statistics to trade-off exploitation (where $\mu_t(\mathbf{x}; \boldsymbol{\theta})$ is high) and exploration (where $\sigma_t^2(\mathbf{x}; \boldsymbol{\theta})$ is high) effectively. Among the various acquisition functions proposed, expected improvement (EI) [23] remains the default choice in many BO applications [43]. Let us define $\mu_{\boldsymbol{\theta}}^+ = \max_{\mathbf{x} \in \mathcal{X}} \mu_t(\mathbf{x}; \boldsymbol{\theta})$ as the best mean value. The EI acquisition function can then be expressed in closed form as:

$$\alpha(\mathbf{x}; \mathcal{D}_t) = \mathbb{E}[\max\{0, f(\mathbf{x}) - \mu_{\boldsymbol{\theta}}^+\}] = \sigma_t(\mathbf{x}; \boldsymbol{\theta})[u\Phi(u) + \phi(u)], \tag{7}$$

where $u = (\mu_t(\mathbf{x}; \boldsymbol{\theta}) - \mu_{\boldsymbol{\theta}}^+)/\sigma_t(\mathbf{x}; \boldsymbol{\theta})$, and $\phi(\cdot)$ and $\Phi(\cdot)$ are the standard normal density and cumulative distribution functions, respectively.

## A.2  Model Selection

We conduct model selection over a discrete, infinite space of kernels $\mathcal{K} = \{k_1, k_2, \ldots\}$. As each kernel comes with its own hyperparameters, we are actually dealing with a space of kernel families. Thus, when referring to a kernel $k$, we consider the whole family over its hyperparameters $\{k_{\boldsymbol{\theta}} \,|\, \boldsymbol{\theta} \in \Theta\}$. Given some model selection criteria $h : \mathcal{K} \to \mathbb{R}$, our goal is to identify the optimal kernel,

$$k^* = \arg\max_{k \in \mathcal{K}} h(k \,|\, \mathcal{D}). \tag{8}$$

A commonly-used criterion for probabilistic models, such as GPs, is the marginal log-likelihood [46],

$$h(k \,|\, \mathcal{D}) = \log p(\mathbf{y} \,|\, \mathbf{X}, k) = \log \int p(\mathbf{y} \,|\, \mathbf{X}, \boldsymbol{\theta}, k)p(\boldsymbol{\theta})d\boldsymbol{\theta}. \tag{9}$$

Unfortunately, the above likelihood is generally intractable for GPs [36], so we resort to the Laplace approximation [35],

$$\log p(\mathbf{y} \,|\, \mathbf{X}, k) \approx \log p(\mathbf{y} \,|\, \mathbf{X}, \hat{\boldsymbol{\theta}}, k) + \log p(\hat{\boldsymbol{\theta}}) - \frac{1}{2}\log \det \boldsymbol{\Sigma}^{-1} + \frac{d_{\boldsymbol{\theta}}}{2}\log 2\pi \tag{10}$$

where $\hat{\boldsymbol{\theta}}$ denotes the maximum a posteriori (MAP) estimate of the hyperparameters with $d_{\boldsymbol{\theta}}$ being its dimension. The term $\boldsymbol{\Sigma}^{-1} = -\nabla^2 \log p(\boldsymbol{\theta} \,|\, \mathcal{D}, k)|_{\boldsymbol{\theta}=\hat{\boldsymbol{\theta}}}$ represents the Hessian matrix evaluated at the MAP estimate. Note that Eq. (10) can be interpreted as rewarding model fit while penalizing model complexity. In this work, we use the Bayesian Information Criterion (BIC) [39], which was also previously employed in [12] and can be seen as an approximation of the Laplace method.

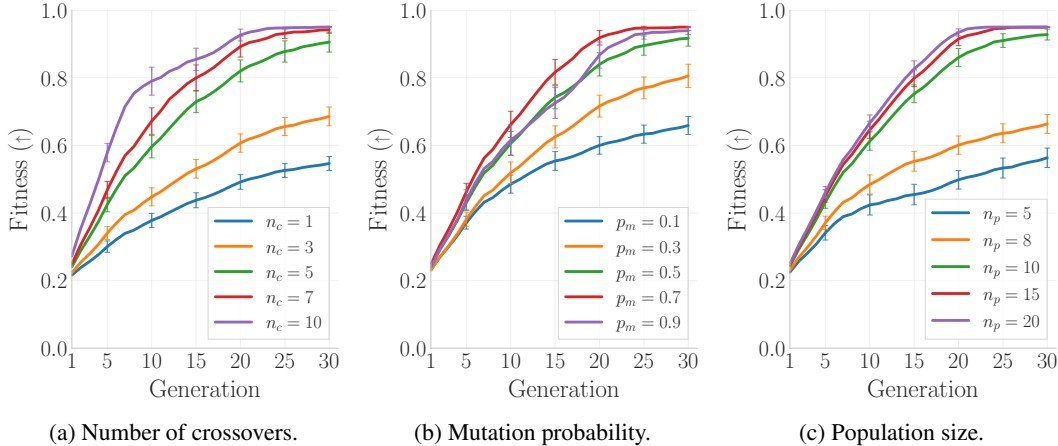

Figure 8: Average fitness (↑) with respect to different CAKE setup. The error bars indicate the standard errors.

# B Experimental Details

In this section, we provide additional details on the implementation, baselines, and benchmarks employed in our experiments.

## B.1 Implementation

As described in Algorithm 1, CAKE depends on three key parameters: the number of crossovers $n_c$, the mutation probability $p_m$, and the population size $n_p$. To evaluate the impact of these parameters, we conduct a sensitivity analysis in Fig. 8, which shows the average fitness of the population over generations under different settings. Below, we provide some intuition for setting these parameters.

**Number of crossovers.** The number of crossovers determines how many new candidate kernels are generated via crossover in each iteration. A higher value of $n_c$ can accelerate the exploration of diverse kernel combinations but increases the number of API calls to the LLM, which may be costly. In our experiments, we found that setting $n_c = 5$ offers a good balance between exploration and efficiency.

**Mutation probability.** Each mutation operation introduces local variations to existing kernels in the population, which may help to refine solutions and escape local optima. The mutation probability controls the likelihood of applying a mutation operation during each iteration. A higher $p_m$ promotes greater diversity and prevents premature convergence, especially when the top-performing kernels become similar. However, excessively high mutation rates risk disrupting promising kernel structures before they can be fully explored. Based on our experiments, we set $p_m = 0.7$ to maintain a balance between sufficient exploration and stable evolution.

**Population size.** The population size dictates how many candidate kernels evolve simultaneously. A larger $n_p$ enhances diversity in the search space and supports more thorough exploration, but also increases computational cost due to repeated GP model fitting for each kernel. Conversely, a smaller $n_p$ may lead to premature convergence or insufficient sampling of the kernel space. Empirically, we found that setting $n_p = 10$ maintains a diverse yet computationally manageable population.

## B.2 Baselines

To test our proposed method, we consider three categories of baseline methods: *fixed*, *adaptive*, and *compositional* approaches.

### B.2.1 Fixed

For the fixed kernel-based baselines, we consider the following widely-used kernels in BO, along with their respective hyperpriors:

1. **Squared Exponential (SE):**

$$k(x, x') = \sigma^2 \exp\left(-\frac{(x - x')^2}{2l^2}\right), \tag{11}$$

where $l$ is the lengthscale parameter and $\sigma$ is the amplitude parameter. The hyperpriors are:

$$l \sim \text{Gamma}(2.0, 2.0), \tag{12a}$$

$$\sigma^2 \sim \text{Gamma}(2.0, 3.0). \tag{12b}$$

2. **Periodic (PER):**

$$k(x, x') = \sigma^2 \exp\left(-\frac{\sin^2\left(\pi|x - x'|/p\right)}{l^2}\right), \tag{13}$$

where $l$ is the lengthscale parameter, $p$ is the period length, and $\sigma$ is the amplitude parameter. The hyperpriors are:

$$l \sim \text{Gamma}(2.0, 2.0), \tag{14a}$$

$$\sigma^2 \sim \text{Gamma}(2.0, 3.0), \tag{14b}$$

$$p \sim \text{Gamma}(2.0, 2.0). \tag{14c}$$

3. **Linear (LIN):**

$$k(x, x') = \sigma^2 xx' + \sigma_c^2, \tag{15}$$

where $\sigma^2$ is the variance parameter, $\sigma_c^2$ is the noise variance. The hyperpriors are:

$$\sigma^2 \sim \text{Gamma}(2.0, 3.0), \tag{16a}$$

$$\sigma_c^2 \sim \text{Gamma}(2.0, 3.0). \tag{16b}$$

4. **Rational Quadratic (RQ):**

$$k(x, x') = \sigma^2 \left(1 + \frac{(x - x')^2}{2\alpha l^2}\right)^{-\alpha}, \tag{17}$$

where $l$ is the lengthscale parameter, $\alpha$ is the relative weighting parameter, and $\sigma$ is the amplitude parameter. The hyperpriors are:

$$l \sim \text{Gamma}(2.0, 2.0), \tag{18a}$$

$$\sigma^2 \sim \text{Gamma}(2.0, 3.0), \tag{18b}$$

$$\alpha \sim \text{Gamma}(2.0, 2.0). \tag{18c}$$

5. **Matérn:**

$$k(x, x') = \sigma^2 \frac{2^{1-\nu}}{\Gamma(\nu)} \left(\sqrt{2\nu}D\right)^\nu K_\nu\left(\sqrt{2\nu}D\right), \tag{19}$$

where

$$D = \frac{(x - x')^2}{l^2} \tag{20}$$

is the distance between $x$ and $x'$ scaled by the lengthscale parameter, $K_\nu$ is the modified Bessel function, and $\sigma$ is the amplitude parameter. In our experiments, the smoothness parameter $\nu$ is set to $1/2$, $3/2$, or $5/2$, corresponding to Matérn-$1/2$ (M1), Matérn-$3/2$ (M3), or Matérn-$5/2$ (M5) respectively. The hyperpriors are:

$$l \sim \text{Gamma}(2.0, 2.0), \tag{21a}$$

$$\sigma^2 \sim \text{Gamma}(2.0, 3.0). \tag{21b}$$

Note that all the above kernels are defined on $\mathbb{R}$ and are applied to input dimension $i$ when indicated by the base kernel symbol, e.g., $\text{SE}_i$ denotes SE kernel is applied to the $i$-th dimension.

Table 2: Details of the OpenML datasets used in the experiments. More information can be found at `https://www.openml.org`.

| Dataset | Task ID | Number of Instances | Number of Features |
|---|---|---|---|
| credit_g | 31 | 1000 | 21 |
| vehicle | 53 | 846 | 19 |
| kc1 | 2109 | 2109 | 22 |
| phoneme | 9952 | 5404 | 6 |
| blood_transfusion | 10101 | 748 | 5 |
| australian | 146818 | 690 | 15 |
| car | 146821 | 1728 | 7 |
| segment | 146822 | 2310 | 20 |
| heart_h | 50 | 294 | 14 |
| tic_tac_toe | 145804 | 958 | 10 |
| kr_vs_kp | 3 | 3196 | 37 |
| qsar | 9957 | 1055 | 42 |

### B.2.2 Adaptive

For the adaptive kernel-based baselines, we adopt the implementation from Roman et al. [37] and apply the following selection criteria:

1. **Random:** This criterion selects a kernel randomly from the set of available kernels.

2. **Utility:** Based on the proposed query points from each kernel, this criterion selects the kernel with the highest utility (acquisition) value:

$$k^* = \arg\max_{k \in \mathbb{K}} \alpha(\mathbf{x}_{t,k}; \mathcal{D}, k). \tag{22}$$

3. **BIC:** This criterion selects the kernel with the lowest BIC value:

$$k^* = \arg\min_{k \in \mathbb{K}} \mathrm{BIC}(k; \mathcal{D}). \tag{23}$$

### B.2.3 Compositional

For the compositional kernel-based baselines, we consider the following methods:

1. **Deep GP.** For the deep GP baseline, we use the `DeepGP` implementation from GPyTorch, where training and inference are conducted using the doubly stochastic variational inference method [38].

2. **Ensemble GP.** For the ensemble GP baseline, we follow the implementation suggested by Lu et al. [31], using the six base kernels used in CAKE to form the kernel dictionary.

3. **Compositional Kernel Search (CKS).** For the CKS baseline, we start from the same base kernels used in CAKE and apply greedy search to search for the kernel structures [12].

4. **Automated BO (ABO)** For the ABO baseline, we use the code provided by the authors at `https://github.com/gustavomalkomes/abo` and follow the setup suggested in [32].

### B.3 Benchmarks

All experiments on the benchmarks were conducted locally on a consumer-grade laptop[4], except for the photonic chip design experiment in Section 5.3, which was executed on a high-performance computing (HPC) cluster due to the computational demands of the physics-based simulation.

### B.3.1 Hyperparameter Optimization

**Datasets.** We include 12 OpenML datasets available in the HPOBench package [14]. The details of the selected datasets are given in Table 2.

---

[4]MacBook Air M2 (2022) with an 8-core CPU, 8-core GPU, 8 GB unified memory, and 256 GB SSD storage.

**Search space.** We follow the search space designated in HPOBench, where we discretize the search space to facilitate efficient tabular lookup operations for various configurations [13]. Each hyperparameter is defined by its type (linear or log scale), along with lower and upper bounds. For example, `[log, 0.001, 1.0]` indicates that the hyperparameter values are sampled on a logarithmic scale between 0.001 and 1.0. In contrast, `[linear, 0.0, 1.0]` implies uniform sampling over the interval [0.0, 1.0]. The search space for each ML model is summarized as follows:

- **Logistic Regression** ($d = 2$)
    - `alpha`: Regularization strength, `[log, 0.001, 1.0]`
    - `eta0`: Initial learning rate, `[log, 0.001, 1.0]`
- **Support Vector Machine (SVM)** ($d = 2$)
    - `C`: Inverse of regularization strength, `[log, 0.01, 10.0]`
    - `gamma`: RBF kernel coefficient, `[log, 0.001, 1.0]`
- **Random Forest** ($d = 4$)
    - `max_depth`: Maximum depth of each tree, `[log, 1, 50]`
    - `max_features`: Number of features to consider when looking for the best split, `[linear, 0.0, 1.0]`
    - `min_samples_leaf`: Minimum number of samples required to be at a leaf node, `[linear, 1, 2]`
    - `min_samples_split`: Minimum number of samples required to split an internal node, `[log, 2, 128]`
- **XGBoost** ($d = 4$)
    - `colsample_bytree`: Fraction of features to use per tree, `[linear, 0.1, 1.0]`
    - `eta`: Learning rate that controls the contribution of each tree to the final prediction, `[log, 0.001, 1.0]`
    - `max_depth`: Maximum depth of a tree, `[log, 1, 50]`
    - `reg_lambda`: $L_2$ regularization term on weights, `[log, 0.1, 10.0]`
- **Multi-Layer Perceptron (MLP)** ($d = 5$)
    - `alpha`: $L_2$ penalty (regularization term) coefficient, `[log, 0.001, 1.0]`
    - `batch_size`: Number of training examples used in one forward/backward pass, `[log, 16, 128]`
    - `depth`: Number of hidden layers in the neural network, `[linear, 1, 3]`
    - `learning_rate_init`: Initial learning rate for weight updates, `[log, 0.001, 1.0]`
    - `width`: Number of neurons in each hidden layer, `[log, 16, 128]`

### B.3.2 Controller Tuning

**Robot pushing.** The reward function is defined as: $f(\mathbf{x}) = -\sum_{i=1}^{2} |\mathbf{x}_{gi} - \mathbf{x}_{si}| - |\mathbf{x}_{gi} - \mathbf{x}_{fi}|$, where $\mathbf{x}_{si}$ represents the starting positions of the objects, $\mathbf{x}_{fi}$ denotes their final positions, and $\mathbf{x}_{gi}$ indicates the goal. The objective is to minimize the total distance from the initial and final positions of the objects to their respective goals, thereby maximizing the reward. We use the original code provided by Wang et al. [51], which is available online at `https://github.com/zi-w/Ensemble-Bayesian-Optimization`.

**Lunar lander.** The reward system includes +100 points for a successful landing, -100 points for a crash, +10 points per frame for each leg in contact with the ground, -0.3 points per frame for firing the main engine, and -0.03 points per frame for firing side engines. We implement the lunar lander environment using the code from `https://github.com/Farama-Foundation/Gymnasium`.

### B.3.3 Photonic Chip Design

**Objective.** The design parameters and indicators for optimizing the photonic chip are detailed in Figure 9. We established target values based on our experiments and a literature review to identify optimal standards for a high-quality chip [28]. For instance, a wavelength of 1310 nm is crucial for telecommunications and satellite applications, while high output power is essential for fields such as

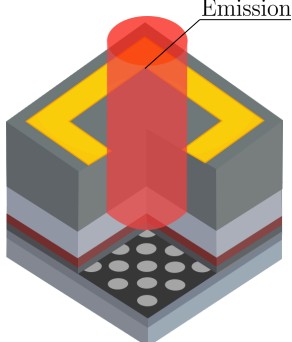

| Design Parameters | |
| --- | --- |
| Length | Lattice constant |
| Width | Size of holes |
| Layer thickness | Filling factor |
| Shape of holes | Material properties |

| Indicator | Target |
| --- | --- |
| Q-factor (Q) | $Q^* = 2 \times 10^6$ |
| Wavelength ($\lambda$) | $\lambda^* = 1310$ |
| Lasing area (A) | $A^* = 3.6 \times 10^{-13}$ |
| Power ($P$) | $P^* = 0.5$ |
| Divergence angle ($\omega$) | $\omega^* = 1.0$ |

Figure 9: Illustration of a photonic chip and its design parameters. The inverse design problem focuses on optimizing these parameters to satisfy certain performance indicators.

Table 3: Details of the test functions used in the experiments.

| Function | Domain | $d$ |
| --- | --- | --- |
| Ackley-$d$ | $[-5, 5]^d$ | 2, 5 |
| Beale | $[-1, 1]^2$ | 2 |
| Branin | $[-5, 10]^2$ | 2 |
| Dropwave | $[-5.12, 5.12]^2$ | 2 |
| Eggholder | $[-512, 512]^2$ | 2 |
| Griewank-$d$ | $[-600, 600]^d$ | 2, 5 |
| Hartmann | $[0, 1]^3$ | 3 |
| Levy | $[-10, 10]^d$ | 2, 3 |
| Rastringin-$d$ | $[-5.12, 5.12]^d$ | 2, 4 |
| Rosenbrock | $[-5, 10]^2$ | 2 |
| Six-Hump Camel | $[-3, 3] \times [-2, 2]$ | 2 |

autonomous driving and medicine. Additionally, a small divergence angle is vital for ensuring high beam quality and effective long-distance light propagation. Based on these performance indicators, we can define the objectives as follows,

$$f_1 = 1 - \frac{Q^* - Q}{Q^*}, \tag{24a}$$

$$f_2 = 1 - \frac{|\lambda^* - \lambda|}{\lambda^*}, \tag{24b}$$

$$f_3 = 1 - \frac{A^* - A}{A^*}, \tag{24c}$$

$$f_4 = 1 - \frac{P^* - P}{P^*}, \tag{24d}$$

$$f_5 = 1 + \frac{\omega^* - \omega}{\omega^*}. \tag{24e}$$

Q-factor ($f_1$) is related to the loss and threshold of the laser, wavelength ($f_2$) is the operation wavelength of the laser, lasing area ($f_3$) is the area of the laser beam at the laser's top surface, power ($f_4$) is the lasing power of the laser in watts, and divergence angle ($f_5$) is the angle between outer boundary and centerline of the laser beam.

## B.4  Prompt Design

Our prompts are designed based on three core components:

- **Problem Context:** The optimization history, including the observed input-output pairs and corresponding fitness scores, provided as in-context examples to guide kernel evolution.

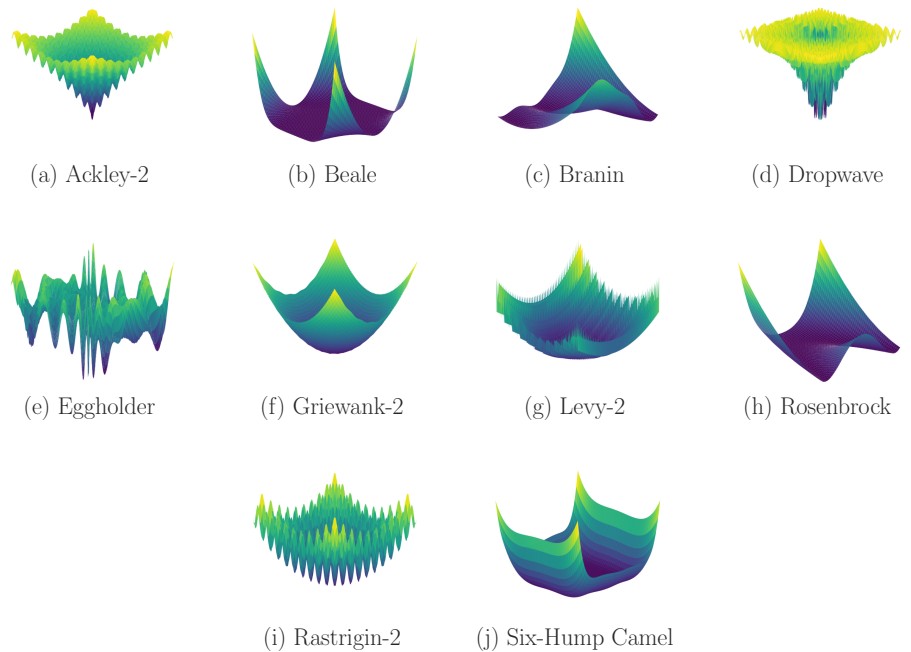

Figure 10: Visualization of the optimization landscapes of two-dimensional test functions.

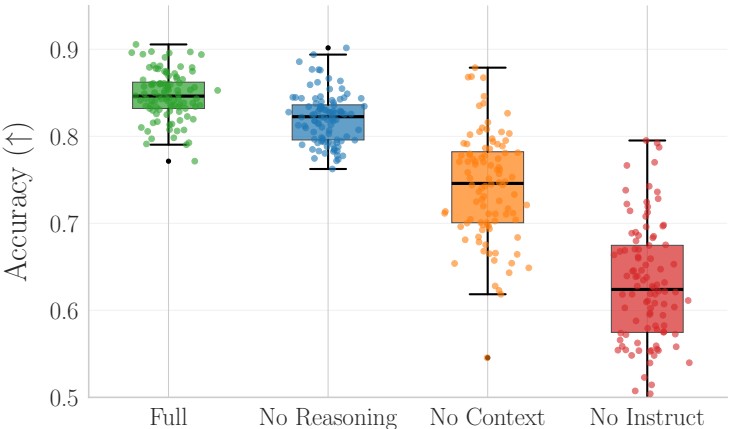

Figure 11: Ablation of prompt components on HPOBench averaged over 20 different random seeds.

- **Task Instruction:** The role assignment and explicit guidelines on how to generate valid kernels using the kernel grammar during crossover and mutation operations.

- **Reasoning:** A phrase asking the LLM to provide a brief natural language explanation for each proposed kernel.

To evaluate the contribution of each component, we conduct an ablation study with the following configurations:

- *Full:* This is the vanilla CAKE setup employed in our experiments, incorporating all three components.

- *No Context:* This variant evaluates the importance of problem context by removing the optimization history (i.e., the observations and fitness values), while keeping the other components.

Table 4: Average normalized regret (↓) over 20 random seeds for different test functions and methods. The values in parentheses indicate the standard errors. The best value is highlighted in **bold** and the second best value is underlined.

| Function | Fixed | | Adaptive | | | Compositional | | | | |
|---|---|---|---|---|---|---|---|---|---|---|
| | SE | M5 | Random | Utility | BIC | DGP | EGP | CKS | ABO | CAKE |
| Ackley-2 | 0.1773 | 0.1220 | 0.1358 | 0.1062 | 0.1863 | 0.2510 | 0.1878 | 0.1012 | 0.1020 | **0.0783** |
| | (0.0232) | (0.0262) | (0.0210) | (0.0165) | (0.0187) | (0.0357) | (0.0196) | (0.0177) | (0.0195) | (0.0203) |
| Ackley-5 | 0.3185 | 0.2369 | **0.1722** | 0.2278 | 0.3590 | 0.3110 | 0.2285 | 0.1812 | 0.1910 | 0.1732 |
| | (0.0117) | (0.0184) | (0.0145) | (0.0171) | (0.0338) | (0.0215) | (0.0139) | (0.0856) | (0.0188) | (0.0250) |
| Beale | 0.3554 | 0.3522 | 0.3855 | 0.4410 | 0.3571 | 0.4775 | 0.3088 | 0.4040 | 0.3118 | **0.2565** |
| | (0.0845) | (0.1006) | (0.0775) | (0.0989) | (0.0806) | (0.0958) | (0.0898) | (0.0712) | (0.0733) | (0.0786) |
| Branin | 0.0183 | 0.0155 | 0.0227 | 0.0372 | 0.0371 | 0.4810 | 0.2045 | 0.0301 | 0.0101 | **0.0070** |
| | (0.0080) | (0.0037) | (0.0106) | (0.0107) | (0.0152) | (0.1017) | (0.0725) | (0.0122) | (0.0478) | (0.0534) |
| Dropwave | 0.5110 | 0.5411 | 0.5460 | 0.5265 | 0.5461 | 0.5560 | 0.6290 | 0.5788 | 0.5529 | **0.4690** |
| | (0.0568) | (0.0523) | (0.0698) | (0.0622) | (0.0750) | (0.0531) | (0.0651) | (0.0669) | (0.0611) | (0.0538) |
| Eggholder | 0.4941 | 0.3545 | 0.4015 | 0.4855 | 0.5485 | 0.4535 | 0.4345 | 0.4536 | 0.4210 | **0.1241** |
| | (0.0602) | (0.0452) | (0.0416) | (0.0527) | (0.0749) | (0.0615) | (0.0497) | (0.0533) | (0.0516) | (0.0541) |
| Griewank-2 | 0.1196 | 0.1282 | 0.1295 | 0.1310 | 0.1272 | 0.1156 | 0.0935 | 0.0589 | 0.0357 | **0.0267** |
| | (0.0692) | (0.0687) | (0.0686) | (0.0685) | (0.0686) | (0.0297) | (0.0248) | (0.0686) | (0.0244) | (0.0256) |
| Griewank-5 | 0.0204 | 0.0223 | 0.0232 | **0.0178** | 0.0281 | 0.0815 | 0.0478 | 0.0258 | 0.0320 | 0.0185 |
| | (0.0032) | (0.0059) | (0.0051) | (0.0033) | (0.0096) | (0.0171) | (0.0110) | (0.0076) | (0.0125) | (0.0133) |
| Hartmann | 0.0007 | 0.0019 | 0.0021 | 0.0358 | 0.6800 | 0.1305 | 0.1780 | **0.0001** | **0.0001** | **0.0001** |
| | (0.0001) | (0.0011) | (0.0009) | (0.0159) | (0.0734) | (0.0441) | (0.0402) | (0.0611) | (0.0544) | (0.0529) |
| Levy-2 | 0.1562 | 0.0418 | 0.0835 | 0.0555 | 0.1145 | 0.1965 | 0.0765 | 0.0668 | 0.0519 | **0.0353** |
| | (0.0684) | (0.0227) | (0.0460) | (0.0060) | (0.0666) | (0.0326) | (0.0198) | (0.0431) | (0.0187) | (0.0197) |
| Levy-3 | 0.1141 | 0.1422 | 0.1495 | 0.0880 | 0.1125 | 0.2265 | 0.0805 | 0.0580 | 0.0590 | **0.0505** |
| | (0.0209) | (0.0403) | (0.0290) | (0.0159) | (0.0230) | (0.0511) | (0.0173) | (0.0194) | (0.0181) | (0.0190) |
| Rastringin-2 | 0.4325 | 0.4251 | 0.5310 | 0.3455 | 0.4490 | 0.5405 | 0.3869 | 0.3722 | 0.3420 | **0.3341** |
| | (0.0571) | (0.0765) | (0.0613) | (0.0442) | (0.0632) | (0.0751) | (0.0338) | (0.0588) | (0.0397) | (0.0468) |
| Rastringin-4 | 0.5765 | 0.5461 | 0.4815 | 0.5905 | 0.5200 | 0.5340 | 0.3270 | 0.3285 | 0.3179 | **0.3128** |
| | (0.0482) | (0.0671) | (0.0400) | (0.0509) | (0.0529) | (0.0475) | (0.0329) | (0.0511) | (0.0375) | (0.0499) |
| Rosenbrock | 0.1025 | 0.0898 | 0.1015 | 0.1405 | 0.1475 | 0.5340 | 0.6040 | 0.0907 | 0.0901 | **0.0483** |
| | (0.0476) | (0.0510) | (0.0573) | (0.0695) | (0.0688) | (0.1101) | (0.0971) | (0.0686) | (0.0907) | (0.0531) |
| Six-Hump Camel | 0.2840 | 0.1507 | 0.3455 | 0.3310 | 0.3265 | 0.4940 | 0.5345 | 0.1071 | **0.1002** | 0.1015 |
| | (0.0856) | (0.0459) | (0.1014) | (0.0760) | (0.0786) | (0.0854) | (0.0528) | (0.0771) | (0.0563) | (0.0669) |
| Mean regret | 0.2454 | 0.2111 | 0.2341 | 0.2373 | 0.3026 | 0.3589 | 0.2881 | 0.1905 | 0.1745 | **0.1357** |
| Median regret | 0.1773 | 0.1422 | 0.1495 | 0.1405 | 0.3265 | 0.4535 | 0.2285 | 0.1012 | 0.1002 | **0.0783** |

- *No Instruct:* This variant excludes task instructions and omits the explicit kernel generation guidelines (i.e., how to perform the crossover and mutation operators), while keeping the other components.

- *No Reasoning:* Thus variant removes the requirement for the LLM to explain its reasoning, producing only the kernel expression.

We evaluate all variants on the HPOBench benchmark, as detailed in Section 5.1. The results, summarized in Figure 11, demonstrate that the full CAKE setup consistently outperforms all ablated variants, underscoring the importance of each prompt component in achieving superior optimization performance. The *No Reasoning* experience a slight degradation in performance, which suggests that the reasoning prompt not only serves to enhance interpretability, but also as a form of self-reflection that helps the LLM evaluate and refine its own proposals. In contrast, the *No Context* variant performs significantly worse, demonstrating that access to optimization history (i.e., observed data points and kernel fitness values) is crucial and that the LLM effectively leverages this information. The *No Instruct* variant performed the worst compared to the other variants. To understand this drop, we examined the percentage of valid kernels, defined as the proportion of kernels that satisfy the kernel properties. We found that the *No Instruct* variant generated valid kernels only 68% of the time, while the other variants consistently produced valid kernels. This is due to the absence of the generation guidelines from the kernel grammar, which underscores its importance in the kernel generation process. Overall, these results strongly support the design choice of our prompts.

# C   Additional Results

In this section, we provide additional results and empirical analysis on our proposed method.

Table 5: Average computational time ($\downarrow$) in seconds per iteration for different methods.

| Method | Time (s) | Main Bottleneck |
|--------|----------|-----------------|
| Fixed | 0.6 | Single GP fitting |
| Adaptive | 3.7 | Multiple GP fitting & kernel selection |
| EGP | 3.9 | Multiple GP fitting & weight update |
| Deep GP | 4.8 | Variational inference |
| CKS | 5.6 | Multiple GP fitting & greedy search |
| ABO | 7.4 | Multiple GP fitting & nested BO loop |
| CAKE | 8.3 | Multiple GP fitting & LLM inference |

## C.1 Benchmark Function Optimization

**Setup.** We consider optimizing a set of test functions commonly used as benchmark for optimization [45]. We provide additional details on the selected test functions[5], including the input domain and dimensionality, in Table 3. We visualize the optimization landscapes of the two-dimensional test functions in Figure 10. From the figure, one can see the challenging nature of these functions, characterized by non-convexity, many local minima, and steep ridges. The goal is to find the global minimum of each test function, where the maximum number of function evaluations is limited to 10 times the dimensionality of the function input, i.e., $T = 10 \times d$. To evaluate the performance of each method, we consider the normalized regret [3]:

$$\frac{f(\mathbf{x}_{\text{opt}}) - f(\mathbf{x}_{\text{best}})}{f(\mathbf{x}_{\text{opt}}) - f(\mathbf{x}_{\text{init}})}, \tag{25}$$

where $f(\mathbf{x}_{\text{init}})$ is the best function value among the initial points, $f(\mathbf{x}_{\text{best}})$ is the best value found by the method, and $f(\mathbf{x}_{\text{opt}})$ is the ground truth optimum. This metric is favorable as it provides a normalized and task-agnostic measure to compare the optimization performance across different tasks.

**Results.** Table 4 shows the normalized regret averaged over 20 seeds for different functions and methods. Our results demonstrate that CAKE outperforms the baselines, achieving roughly a 22.2% improvement in the mean regret and a 21.9% improvement in the median regret compared to the second-best method. Notably, CAKE ranks among the top two for all functions and achieves the best performance 12 times out of 15. Our results also reveal that fixed kernels such as SE and M5, which are the default in BO, are not universally effective and actually perform poorly in many test functions. We also found that adaptive methods exhibit inconsistent performance and generally underperform compared to the compositional approaches like ABO.

## C.2 Computational Time

Table 5 shows the average clock time per iteration for different BO methods. Note that we only measure the time spent on surrogate model computation, not including the black-box function evaluations. In CAKE, we use OpenAI's `gpt-4o-mini` as the LLM. Based on our measurements, it processes about 104.5 tokens per second. Each LLM call, including both the input prompt and output response, uses around 180 tokens. This results in an inference time of approximately $180/104.5 = 1.72$ seconds per call. Each iteration of CAKE also involves fitting GP models for all kernels in the current population. Our experiments show that fitting one GP takes about 0.5 seconds. Therefore, fitting multiple GPs per iteration adds about 3–5 seconds to the total runtime. Other adaptive and compositional kernel methods that maintain multiple GP models also experience similar computational costs. However, methods like CKS and ABO involve additional steps, such as greedy search or nested BO loops, for constructing the kernels, which further increases their runtime. DGP requires performing doubly stochastic variational inference across deep layers, which adds even more computational overhead. In contrast, fixed and simple adaptive kernel methods, which either use a single GP or choose from a small set of kernels, are much faster per iteration. While CAKE has a higher average runtime per iteration than the baselines, this increase is largely due to external

---

[5]The analytic expression as well as the global optimum of these functions can be found at `https://www.sfu.ca/~ssurjano/optimization.html`.

```
┌─────────────────────────────────────────────────────────────────────────────────┐
│ ■ Crossover Response                                                              │
│                                                                                   │
│   Prompt:                                                                         │
│   You are given two parent kernels and their fitness scores: LIN (0.53), SE       │
│   (0.67). Please propose a new kernel that has a potentially higher fitness       │
│   score. You may combine the parent kernels using any of the operators from       │
│   {+, ×}. Briefly explain your reasoning behind the proposed kernel.              │
│   - - - - - - - - - - - - - - - - - - - - - - - - - - - - - - - - - - - - - - -   │
│   Response:                                                                       │
│   Kernel: LIN + SE                                                                │
│   Analysis: The LIN kernel captures linear trends in the data, which may be       │
│   present given the varying function values across different inputs. The SE       │
│   kernel, on the other hand, is effective in modeling smooth variations in the    │
│   data, which could account for the fluctuations observed in the function values  │
│   despite the overall linear trend. By combining these two kernels with the       │
│   addition operator (+), we can leverage the linear trend from the LIN kernel     │
│   while also allowing for smooth deviations from this trend through the SE kernel. │
└─────────────────────────────────────────────────────────────────────────────────┘
```

Figure 12: Sample response for the crossover step.

```
┌─────────────────────────────────────────────────────────────────────────────────┐
│ ■ Mutation Response                                                               │
│                                                                                   │
│   Prompt:                                                                         │
│   You are given a kernel and its fitness score: LIN + SE (0.72). Please propose   │
│   a new kernel that has a potentially higher fitness score. You may replace a     │
│   base kernel in the current expression with another base kernel from the set     │
│   {SE, PER, LIN, RQ, M3, M5}. Briefly explain your reasoning behind the           │
│   proposed kernel.                                                                │
│   - - - - - - - - - - - - - - - - - - - - - - - - - - - - - - - - - - - - - - -   │
│   Response:                                                                       │
│   Kernel: LIN + RQ                                                                │
│   Analysis: The LIN + SE kernel captures a linear trend with smooth variations.   │
│   However, the SE kernel assumes a constant level of smoothness, which may not    │
│   be appropriate for all data patterns. By replacing the SE kernel with the RQ    │
│   kernel, we can introduce varying smoothness, allowing the model to adapt to     │
│   different levels of noise in the data.                                          │
└─────────────────────────────────────────────────────────────────────────────────┘
```

Figure 13: Sample response for the mutation step.

factors such as internet connectivity and API latency. Importantly, in most BO tasks, evaluating the black-box function is far more time-consuming than the BO process itself. Thus, reducing the number of function evaluations (i.e., improving sample efficiency) is often more important than minimizing the wall-clock time per iteration.

## C.3 Case Study

We conducted a case study to examine the LLM's outputs and assess whether it provides useful responses. Specifically, we analyzed sample responses from the LLM during the crossover and mutation steps, which are shown in Figure 12 and Figure 13, respectively. In Figure 12, the LLM is given two parent kernels, LIN and SE, along with their fitness scores. It suggests combining them using the addition operator (+) to improve the fitness. The explanation states that the LIN kernel captures linear trends, while the SE kernel models smoothness. Thus, combining them as LIN + SE allows the new kernel to represent both linear and smooth patterns in the data. In Figure 13, the LLM is given the kernel LIN + SE and its fitness score. It proposes replacing the SE kernel with the RQ kernel to further improve fitness. The reasoning is that while the SE kernel assumes constant smoothness, the RQ kernel allows for varying levels of smoothness. This makes the model more flexible, especially when dealing with data that has changing noise patterns. These examples show that the LLM posesses a solid understanding about the kernel properties and how they affect modeling (i.e., how different kernels can be combined or modified to better fit specific data characteristics), enabling it to make meaningful proposals during the kernel generation process.

## C.4 Interpretability of CAKE

By design, the kernel grammar used in CAKE enables us to automatically generate interpretable descriptions of the data based on the proposed hypothesis (kernel expression). In this section, we analyze the kernel expression discovered by CAKE for one of the hyperparameter optimization tasks from HPOBench, which involves tuning an SVM model on the `credit_g` dataset. Recall that, for the SVM model, we have two hyperparameters to tune: the regularization parameter `C` and the RBF kernel coefficient `gamma`. The kernel expression proposed by CAKE for this specific task is,

$$(\text{SE} \times \text{PER}) + (\text{LIN} \times \text{RQ}) + (\text{M5} \times \text{SE}). \tag{26}$$

Existing kernel grammar-based methods, such as Automatic Bayesian Covariance Discovery (ABCD) [30], typically interpret such expressions by first simplifying them into a canonical sum-of-products form to enable modular interpretation. The natural language descriptions are then generated by applying predefined template rules that map each kernel component and its parameters to descriptive phrases. In contrast, in our CAKE framework, the LLM itself acts as the interpreter of the kernel structures, producing natural language descriptions directly and automatically "for free" without the need for explicit simplification or handcrafted template rules (see Fig. 14. This level of interpretability highlights a key advantage of our CAKE framework. The LLM's contextual understanding and access to broad domain knowledge enable it not only to evolve expressive kernels, but also to explain them through meaningful natural language descriptions. Unlike traditional methods that treat kernel learning as a purely numerical optimization task, CAKE generates interpretable hypotheses that reveal insights into the structure of the underlying "black-box" objective function.

---

**CAKE's Interpretation**

**Response:**
Kernel: $(\text{SE} \times \text{PER}) + (\text{LIN} \times \text{RQ}) + (\text{SE} \times \text{M5})$
Analysis: The observations can be described using a combination of three distinct patterns. The first term, $\text{SE} \times \text{PER}$, captures recurring fluctuations in the performance across the (`C`, `gamma`) space. High `gamma` values lead to overfitting, but increasing `C` mitigates this effect, resulting in a overfitting-recovery cycle. The second term, $\text{LIN} \times \text{RQ}$, models a general increase in performance as `C` increases, as stronger regularization improves generalization. However, the RQ component allows the model to adapt to varying levels of smoothness which is useful near extreme `gamma` values. The third term, $\text{M5} \times \text{SE}$, combines the M5's robustness to non-smooth changes with SE's smoothness to capture complex, localized variations in performance that are not easily explained by global trends or periodicity alone, such as sharp drops in accuracy when both `C` and `gamma` are too large.

---

Figure 14: Intepretation for the SVM hyperparameter optimization task on the `credit_g` dataset.

## C.5 Choice of LLM

In this section, we analyze the impact of the LLM choice on the performance of our method. We evaluated three open-source models—Qwen's `qwen-2.5-7b`, DeepSeek's `deepseek-r1-distill-qwen-7b`, and Meta's `llama-3.1-8b`—alongside three closed-source models: Google's `gemini-2.0-flash`, Anthropic's `claude-3.5-haiku`, and OpenAI's `gpt-4o-mini`, on the HPOBench functions described in Section 5.1. The results in Table 6 show that, despite some variations in performance across different HPOBench functions, there is a clear trend: our method performs better when using more recent and capable LLMs. We also observe a performance boost with `deepseek-r1-distill-qwen-7b`, suggesting that reasoning-based models can further improve results. However, this gain comes at the cost of longer inference time, a trade-off we plan to explore in future work. Overall, these results indicate that as LLMs become more advanced, our method naturally benefits from their improved capabilities.

## C.6 Choice of Acquisition Function

We test the robustness of CAKE under different choice of acquisition functions, including EI [23], upper confidence bound (UCB) [44], and Thompson sampling (TS) [47]. While each acquisition function embodies a different exploration-exploitation trade-off (i.e., TS exhibits slightly higher standard errors due to its stochastic nature while CAKE-UCB occasionally underperforms or outperforms

Table 6: Average accuracy (↑) on HPOBench over 20 random seeds for different LLMs. The values in parentheses indicate the standard errors.

| LLM | LR | SVM | RF | XGB | MLP |
|---|---|---|---|---|---|
| qwen-2.5-7b | 0.7720 (0.0310) | 0.8520 (0.0180) | 0.8810 (0.0260) | 0.9030 (0.0380) | 0.8610 (0.0330) |
| deepseek-r1-distill-qwen-7b | 0.8122 (0.0210) | 0.8630 (0.0141) | 0.8964 (0.0234) | 0.9211 (0.0320) | 0.8692 (0.0292) |
| llama-3.1-8b | 0.7815 (0.0290) | 0.8581 (0.0177) | 0.8873 (0.0252) | 0.9110 (0.0355) | 0.8652 (0.0312) |
| claude-3.5-haiku | 0.8177 (0.0231) | 0.8655 (0.0130) | 0.8982 (0.0242) | 0.9240 (0.0331) | 0.8712 (0.0301) |
| gemini-2.0-flash | 0.8253 (0.0204) | 0.8720 (0.0121) | 0.9056 (0.0223) | 0.9310 (0.0303) | 0.8780 (0.0281) |
| gpt-4o-mini | 0.8188 (0.0220) | 0.8663 (0.0130) | 0.8991 (0.0250) | 0.9257 (0.0340) | 0.8722 (0.0310) |

Table 7: Average normalized regret (↓) $\pm$ standard error over 20 random seeds for CAKE with different acquisition functions. Value that are not significantly different from the lowest average regret for each function are **bolded**.

| Function | CAKE-EI | CAKE-UCB | CAKE-TS |
|---|---|---|---|
| Ackley-2 | **0.0783** $\pm$ 0.0203 | **0.0812** $\pm$ 0.0215 | 0.0921 $\pm$ 0.0287 |
| Ackley-5 | **0.1732** $\pm$ 0.0250 | 0.1654 $\pm$ 0.0262 | **0.1789** $\pm$ 0.0310 |
| Beale | **0.2565** $\pm$ 0.0786 | **0.2488** $\pm$ 0.0810 | 0.2720 $\pm$ 0.0920 |
| Branin | **0.0070** $\pm$ 0.0534 | **0.0065** $\pm$ 0.0510 | 0.0082 $\pm$ 0.0601 |
| Dropwave | 0.4690 $\pm$ 0.0538 | 0.4820 $\pm$ 0.0560 | 0.5010 $\pm$ 0.0650 |
| Egg holder | **0.1241** $\pm$ 0.0541 | 0.1350 $\pm$ 0.0570 | **0.1298** $\pm$ 0.0620 |
| Griewank-2 | **0.0267** $\pm$ 0.0256 | **0.0275** $\pm$ 0.0260 | 0.0310 $\pm$ 0.0305 |
| Griewank-5 | **0.0185** $\pm$ 0.0133 | **0.0190** $\pm$ 0.0140 | **0.0195** $\pm$ 0.0180 |
| Hartmann | **0.0001** $\pm$ 0.0529 | **0.0003** $\pm$ 0.0515 | **0.0005** $\pm$ 0.0580 |
| Levy-2 | **0.0353** $\pm$ 0.0197 | **0.0360** $\pm$ 0.0205 | 0.0402 $\pm$ 0.0240 |
| Levy-3 | **0.0505** $\pm$ 0.0190 | **0.0580** $\pm$ 0.0200 | **0.0520** $\pm$ 0.0235 |
| Rastrigin-2 | 0.3341 $\pm$ 0.0468 | 0.3520 $\pm$ 0.0490 | 0.3650 $\pm$ 0.0580 |
| Rastrigin-4 | **0.3128** $\pm$ 0.0499 | **0.3150** $\pm$ 0.0510 | 0.3300 $\pm$ 0.0570 |
| Rosenbrock | **0.0483** $\pm$ 0.0531 | **0.0490** $\pm$ 0.0540 | **0.0510** $\pm$ 0.0600 |
| Six-Hump Camel | **0.1015** $\pm$ 0.0669 | **0.0920** $\pm$ 0.0680 | **0.1030** $\pm$ 0.0750 |

EI due to its optimism bias), CAKE consistently achieves comparable performance across all variants. As shown in table below, the performance of CAKE-UCB and CAKE-TS remains close to that of CAKE-EI across a diverse set of benchmark functions. To rigorously assess statistical equivalence, we conducted a one-sided paired Wilcoxon signed-rank test (at the 5% significance level) comparing CAKE-EI against CAKE-UCB and CAKE-TS across multiple random seeds. The results indicate that, on the majority of functions, differences are not statistically significant. Overall, these results confirm that CAKE's effectiveness is not sensitive to the specific choice of acquisition function.

## C.7 Quantitative Analysis

To rigorously support our claim in Section 5.1 that CAKE excels in the early stages of optimization, we conducted a quantitative analysis based on *normalized improvement*, defined as:

$$\text{Normalized improvement at trial } t = \frac{f_t - f_0}{f^* - f_0}, \tag{27}$$

where $f_0$ is the initial performance, $f_t$ is the best performance at trial $t$, and $f^*$ is the final performance at 100% budget. This measures how much of the total progress CAKE achieves up to the $t$-th iteration. As shown in Table 8, CAKE achieves 67.5% of its total improvement within just 25% of the budget, on average. By 50%, it reaches over 83% of its final improvement, and by 75%, it is nearly converged.

Table 8: Normalized improvement on HPOBench at different budget level.

| Budget | 25% | 50% | 75% |
|---|---|---|---|
| LR | 0.6183 | 0.8065 | 0.8401 |
| SVM | 0.7007 | 0.8244 | 0.9481 |
| RF | 0.6244 | 0.7685 | 0.8646 |
| XGB | 0.6914 | 0.8643 | 0.9767 |
| MLP | 0.7394 | 0.9155 | 0.9742 |
| Average | 0.6749 | 0.8358 | 0.9207 |

This demonstrates that CAKE rapidly identifies effective kernels and drives fast early progress, making it particularly effective in data-scarce regimes.

## D    Limitations and Future Work

**Computational cost.** While CAKE can be applied entirely in-context and does not require any fine-tuning, using LLMs for inference may result in a larger computational footprint compared to traditional BO methods (see Appendix C.2). Despite this, our findings indicate that CAKE trades this off with improved sample efficiency, which is a particularly desirable property for black-box optimization tasks. This suggests the potential for integrating CAKE with more computationally efficient approaches, such as deploying it in the earlier stage of the optimization process.

**Data Contamination.** We acknowledge the possibility that LLMs may have been exposed to scientific literature or code related to common optimization benchmarks during pre-training. However, we argue that data contamination is unlikely to meaningfully affect our results. While the LLM may possess general knowledge about kernels or synthetic functions, our approach to adaptive kernel evolution in BO is novel and there is no evidence that the specific kernel expressions or the optimization trajectory, exist in any public dataset or text. Thus, we believe that the observed performance stems from in-context adaptation, not memorization. This is further supported by our ablation study in Section B.4 which shows that removing the observed data from the prompt leads to significant performance degradation, confirming that the LLM relies on in-context learning rather than prior knowledge alone.

**Generalized kernel grammar.** While we focus on addition and multiplication as initial proof-of-concept operators, these operators are in fact good enough to form a rich and expressive space of kernels. For example, by only using these operations, we can construct polynomial kernels to capture non-linear patterns as well as multi-dimensional kernels to model interactions among input features [12]. However, we would like that to note that the kernel grammar can be extended using other operators that preserve the closure properties of kernel functions, such as convolution, composition, and affine transformations [42]. We aim to explore these possibilities further in a future work.

**Extension to broader ML tasks.** Our long-term goal is to develop a universal adaptive kernel method that can be applied across a wide range of ML tasks. While the current work focuses on BO, the underlying idea of using an LLM to guide kernel evolution is not task-specific. We believe that CAKE can be easily adapted for other kernel-based methods such as SVM-based regression and classification, kernel principal component analysis, and metric learning with kernels. By leveraging task-specific performance signals, CAKE can automate and enhance kernel design across a variety of kernel-based methods, demonstrating its broader potential to improve ML applications.

