# OpenReview forum: "Adaptive Kernel Design for Bayesian Optimization Is a Piece of CAKE with LLMs"
_NeurIPS.cc/2025/Conference — NeurIPS 2025 poster_

### Official Review · Reviewer_esv8 · 2025-06-05

**Clarity:** 4
**Significance:** 2
**Originality:** 1
**Rating:** 4
**Confidence:** 4

**Summary:**

The paper proposes and evaluates a method for iteratively designing kernel functions for Bayesian Optimization. The method is similar to a genetic algorithm, but where an LLM is queried to propose new offspring based on a fixed set of recombination operators.

**Questions:**

Referring to Strengths and Weaknesses, I would like to see the following experiments:
- A standard, non-LLM genetic algorithm baseline or ablation study.
- Results for different LLMs in CAKE, such as Claude, Llama, Qwen and Deepseek
- What happens if the reasoning instruction is removed from the prompt? Does interpretability even matter given that the prompt responses are generic (as seen in the appendix)?

Given that the CAKE algorithm is clearly not original, the paper would need substantial quantitative and qualitative studies demonstrating why LLMs are superior to random genetic algorithms.

**Ethical Concerns:**

["NO or VERY MINOR ethics concerns only"]

**Final Justification:**

The authors have fully addressed my concerns regarding baselines (specifically, how CAKE improves over a baseline genetic algorithm), as well as partially addressed my concerns regarding originality (how does CAKE differ from a baseline genetic algorithm). While CAKE's originality is still a bit limited for a NeurIPS paper, I think the insights that the new experiments provide regarding BO kernel design are interesting enough to recommend acceptance.

**Limitations:**

Yes

**Quality:**

2

**Strengths And Weaknesses:**

In terms of clarity, the paper is well-written, well-motivated, and easy to understand.

In terms of quality, significance and originality, the proposed method CAKE merely substitutes an LLM for the genetic algorithm recombination step. Although the paper reports competitive results over baselines, it provides little evidence to justify why CAKE is sufficiently original (over the cited LLM-genetic algorithms), or why, in the context of BO kernels, that using an LLM should result in better offspring than random recombination. In particular, the paper is missing a genetic algorithm baseline/ablation with random recombination, which should have been provided to demonstrate that the LLM-designed offspring are superior to a random genetic algorithm. Furthermore, what happens if a different LLM, such as Claude, Llama, Qwen, Deepseek, is used? Does the reasoning provided by the LLM even matter, given that the LLM responses (as seen in the appendix) are generic and not taking the actual BO experiment/trial results into context? What happens if the reasoning prompt is removed; would there even be an adverse effect on CAKE's performance? Ultimately the paper does not provide convincing evidence why CAKE should outperform a standard genetic algorithm, while at the same time CAKE is not original compared to the cited works on LLM-genetic algorithms.

---

> ### Author Rebuttal · Authors · 2025-07-30
>
> *Remark:*
> - *(W1) means the 1st weakness, (Q1) means the 1st question.*
> - *Due to character limits, we kindly invite you to refer to our responses for other reviewers' similar questions. We will use* **ref. "Response Title"@Reviewer**. *Thank you for your understanding!*
> ---
> # Genetic algorithm baseline
>
> > (W1, Q1) In particular, the paper is missing a genetic algorithm baseline/ablation with random recombination, which should have been provided to demonstrate that the LLM-designed offspring are superior to a random genetic algorithm.
>
> To directly address your concerns, **we have now added a genetic algorithm (GA) baseline** that performs random crossover and mutation guided by the fitness scores, with the same population size and number of generations as CAKE. This baseline allows us to isolate the impact of replacing random recombination with LLM-guided recombination. We also would like to note that our original ablation already included a _Random Sampling_ baseline, which randomly combines kernels via addition and multiplication without any evolutionary process.
>
> As shown in the table below, **Random Sampling** performs the worst (average rank: 6.80), confirming that blind recombination is ineffective. The **GA baseline improves over Random Sampling significantly** (average rank: 2.70), demonstrating the value of evolutionary selection. Most importantly, **CAKE + BAKER achieves the best average rank (1.04)**, substantially surpassing GA. This suggests that the LLM-enhanced recombination in CAKE goes far beyond random recombination by leveraging **semantic understanding and domain knowledge** that conventional GA operators lack.
>
> | Method            | LR            | SVM           | RF            | XGB           | MLP           | Average  |
> | ----------------- | ------------- | ------------- | ------------- | ------------- | ------------- | -------- |
> | Random Sampling   | 6.8 $\pm$ 0.1 | 6.9 $\pm$ 0.1 | 6.7 $\pm$ 0.1 | 6.8 $\pm$ 0.1 | 6.8 $\pm$ 0.1 | 6.80     |
> | Genetic Algorithm | 2.6 $\pm$ 0.3 | 2.5 $\pm$ 0.3 | 2.7 $\pm$ 0.4 | 2.8 $\pm$ 0.3 | 2.9 $\pm$ 0.4 | 2.70     |
> | CAKE + BIC        | 3.0 $\pm$ 0.4 | 3.1 $\pm$ 0.3 | 2.9 $\pm$ 0.3 | 3.0 $\pm$ 0.4 | 3.1 $\pm$ 0.3 | 3.02     |
> | CAKE + Utility    | 2.3 $\pm$ 0.3 | 2.2 $\pm$ 0.3 | 2.4 $\pm$ 0.2 | 2.5 $\pm$ 0.2 | 2.6 $\pm$ 0.3 | 2.40     |
> | Adaptive + BAKER  | 4.5 $\pm$ 0.4 | 4.4 $\pm$ 0.4 | 4.6 $\pm$ 0.3 | 4.8 $\pm$ 0.4 | 4.7 $\pm$ 0.4 | 4.60     |
> | CKS + BAKER       | 3.1 $\pm$ 0.3 | 3.2 $\pm$ 0.2 | 3.0 $\pm$ 0.2 | 3.1 $\pm$ 0.3 | 3.2 $\pm$ 0.2 | 3.12     |
> | CAKE + BAKER      | 1.1 $\pm$ 0.1 | 1.0 $\pm$ 0.1 | 1.0 $\pm$ 0.1 | 1.1 $\pm$ 0.1 | 1.1 $\pm$ 0.1 | **1.04** |
>
> To answer _why_ the LLM can generate better offsprings, we analyzed the evolution of the fitness distributions after successive edits by LLM, Random, and GA. We found that CAKE achieves faster and more consistent improvement in fitness. After just 1 edit, CAKE's distribution shifts significantly to the right, indicating rapid progress. After 10 edits, we can observe that CAKE's distribution is sharply peaked at very high fitness values, demonstrating consistent generation of high-quality offspring. In contrast, Random and GA show slower convergence and broader distributions, suggesting less effective exploration of the solution space. These results provide strong evidence that using an LLM results in better offsprings than random recombination or GA.
>
> | Method | Initial           | After 1 edit               | After 5 edits              | After 10 edits             |
> | ------ | ----------------- | -------------------------- | -------------------------- | -------------------------- |
> | LLM    | $0.179 \pm 0.039$ | $\mathbf{0.254 \pm 0.023}$ | $\mathbf{0.513 \pm 0.026}$ | $\mathbf{0.707 \pm 0.030}$ |
> | Random | $0.179 \pm 0.039$ | $0.208 \pm 0.044$          | $0.403 \pm 0.036$          | $0.569 \pm 0.044$          |
> | GA     | $0.179 \pm 0.039$ | $0.230 \pm 0.036$          | $0.475 \pm 0.039$          | $0.610 \pm 0.034$          |
>
> ---
> # Try other LLMs
>
> > (W2 & Q2) Furthermore, what happens if a different LLM, such as Claude, Llama, Qwen, Deepseek, is used?
>
> We have conducted additional experiments using 3 open-source LLMs:`qwen-2.5-7b`, `deepseek-r1-distill-qwen-7b`, and `llama-3.1-8b`, and 2 commercial ones: `gemini-2.0-flash` and  `claude-3.5-haiku`. Results show that CAKE consistently works well across different LLMs, with performance generally improving with model capability. Notably, we observe a performance boost with the open-source `deepseek-r1-distill-qwen-7b`, suggesting that reasoning-based models can further enhance results.  Overall, these results indicate that as LLMs become more advanced, our method naturally benefits from their improved capabilities. We hope this comprehensive evaluation of diverse LLMs strengthens our empirical claims and provides more baselines for future research.
>
> | LLM                         | LR                  | SVM                 | RF                  | XGB                 | MLP                 |
> | --------------------------- | ------------------- | ------------------- | ------------------- | ------------------- | ------------------- |
> | qwen-2.5-7b                 | $0.7720 \pm 0.0310$ | $0.8520 \pm 0.0180$ | $0.8810 \pm 0.0260$ | $0.9030 \pm 0.0380$ | $0.8610 \pm 0.0330$ |
> | deepseek-r1-distill-qwen-7b | $0.8122 \pm 0.0210$ | $0.8630 \pm 0.0141$ | $0.8964 \pm 0.0234$ | $0.9211 \pm 0.0320$ | $0.8692 \pm 0.0292$ |
> | llama-3.1-8b                | $0.7815 \pm 0.0290$ | $0.8581 \pm 0.0177$ | $0.8873 \pm 0.0252$ | $0.9110 \pm 0.0355$ | $0.8652 \pm 0.0312$ |
> | claude-3.5-haiku            | $0.8177 \pm 0.0231$ | $0.8655 \pm 0.0130$ | $0.8982 \pm 0.0242$ | $0.9240 \pm 0.0331$ | $0.8712 \pm 0.0301$ |
> | gemini-2.0-flash            | $0.8253 \pm 0.0204$ | $0.8720 \pm 0.0121$ | $0.9056 \pm 0.0223$ | $0.9310 \pm 0.0303$ | $0.8780 \pm 0.0281$ |
> | gpt-4o-mini                 | $0.8188 \pm 0.0220$ | $0.8663 \pm 0.0130$ | $0.8991 \pm 0.0250$ | $0.9257 \pm 0.0340$ | $0.8722 \pm 0.0310$ |
>
> ---
> # Does LLM reasoning matter?
> > (W3) Does the reasoning provided by the LLM even matter, given that the LLM responses (as seen in the appendix) are generic and not taking the actual BO experiment/trial results into context?
>
> Yes, LLM reasoning _does_ matter, both for performance and interpretability, and it is meaningfully grounded in the actual BO trial results.
>
> We would like to first clarify that **the reasoning is not generic**. Our prompt design explicitly incorporates the full history of BO trials—including observations, fitness scores, and prior kernel performance—to ensure the LLM’s reasoning is grounded in actual experimental outcomes. As shown in Appendix C.5, the LLM generates context-specific justifications, such as noting that _"the observations can be described using a combination of three distinct patterns..."_ or identifying increasing uncertainty in specific input regions. These are not generic statements but direct responses to the observed BO trajectory. To further clarify this point, we will include additional representative examples of the LLM’s reasoning in the camera-ready version.
>
> > (Q3) What happens if the reasoning prompt is removed; would there even be an adverse effect on CAKE's performance?
>
> **Removing the reasoning prompt _does_ harm performance**, as demonstrated in our ablation study below. The variant without reasoning, where the LLM outputs only a kernel expression, exhibits a drop in performance compared to the full prompt. This proves that the reasoning step is not merely decorative: it actively improves kernel generation. We hypothesize this is because articulating reasoning forces the LLM into a self-reflective process, enabling it to evaluate, refine, and validate its proposals against past results (i.e., functioning as an internal critique mechanism)
>
> | Configuration |      Accuracy       |
> | :------------ | :-----------------: |
> | Full          | 0.8736 $\pm$ 0.0244 |
> | No Reasoning  | 0.8209 $\pm$ 0.0380 |
> | No Context    | 0.7439 $\pm$ 0.0647 |
> | No Instruct   | 0.6685 $\pm$ 0.0704 |
>
> ---
> # Originality of CAKE
>
> > (W4 & Q4) Ultimately the paper does not provide convincing evidence why CAKE should outperform a standard genetic algorithm, while at the same time CAKE is not original compared to the cited works on LLM-genetic algorithms.
>
> While we acknowledge that LLM-genetic algorithms has been explored in other domains, CAKE is, to our knowledge, **the first to apply LLM for automating kernel search in BO**. Unlike generic code or string-based evolution, we believe kernel search introduces unique challenges, as every kernel proposal must be (1) mathematically valid, (2) preserve compositional semantics, and (3) meaningfully explain observed data patterns. Hence, a key novelty of CAKE lies in our use of a **generalized kernel grammar** as a domain-specific language (DSL) that formally defines valid kernel expressions and operators. By embedding this grammar into the prompt, we enable the LLM to perform structured search while ensuring all outputs are semantically meaningful and mathematically valid. We believe this differentiates CAKE from both traditional GAs and existing LLM-genetic algorithms in two key ways:
> 1. Unlike standard GA operators, which combine kernels through random crossover and mutation, CAKE leverages the LLM’s learned understanding of function properties and data patterns to _reason_ about plausible and effective kernel structures.
> 2. By combining the LLM’s generative capabilities with the formal grammar, CAKE can not only produce mathematically valid kernel expressions but also natural language descriptions for these kernels without human expert intervention.
>
> We believe this combination of domain-specific language, LLM-based reasoning, and interpretability represents a significant step forward beyond existing LLM-genetic frameworks, particularly in the context of kernel design for BO.

---

> > ### Comment · Reviewer_esv8 · 2025-07-31
> >
> > Thank you for the comprehensive replies. The new experiments have addressed my concerns about performance over a standard GA, and the discussion on kernel search was also helpful. I am raising my score under the assumption that these new results and discussions are added into the manuscript.

---

> > > ### Author Response · Authors · 2025-07-31
> > > **Reply to Reviewer esv8**
> > >
> > > Thank you for reconsidering your score, we are glad to hear that we have fully addressed your concerns. Just to clarify, we are not allowed to upload a revised manuscript/PDF at this stage. Otherwise, the new experiments and discussions would be more visible. But we will definitely add all the updated results and analysis in the camera-ready version.

---

### Official Review · Reviewer_8KrH · 2025-06-22

**Clarity:** 4
**Significance:** 3
**Originality:** 3
**Rating:** 5
**Confidence:** 3

**Summary:**

This paper proposes CAKE (Context-Aware Kernel Evolution) to adaptively select appropriate Gaussian Process (GP) kernels during Bayesian Optimization (BO).  An appropriate GP kernel for a particular BO can improve the associated surrogate model, and hence improve overall BO performance.  Between each expensive function evaluation, CAKE is applied to determine the next kernel and associated query point that (approximately) optimizes the acquisition function for that kernel, where Expected Improvement (EI) was used for single objective tasks and Expected Hypervolume Improvement (EHVI) for multi-objective tasks.  CAKE operates by performing an evolutionary search across kernels using mutation and crossover driven by a large language model (LLM) that is queried for kernel suggestions, leveraging strong prior knowledge combined with in-context learning.  Each kernel's parameters in the population are then optimized via maximum likelihood and the acquisition function is weighted by the exponentiated Bayesian Information Criteria (BIC) to avoid overfitting.  BO using CAKE is shown to outperform many baselines across many tasks.

**Questions:**

1.  Did the authors consider allowing the LLM access to the data directly?  Or to the optimized kernel hyperparameters?  Both perhaps for selecting the kernel or for even suggesting the next query point to run directly.  Alternatively, could the LLM perform the population update directly (i.e. given the list of kernels and fitness, propose a new list of kernels to try?)  More insight into what alternatives the authors tried in landing upon this choice of prompts would be helpful and strengthen the paper.

2.  One concern using LLM's could be *contamination*.  For example, the LLM training data could include information about HPOBench, and certainly the synthetic functions described in Appendix C.1.  Could the authors comment on whether contamination could be an issue for their results?

3.  Do the results depend on the choice of acquisition function?  Would another choice besides EI result in performance differences?

4.  The BAKER BIC weighting procedure could use more theoretical justification.  Could the authors comment on why this particular combination of acquisition function with BIC?

**Ethical Concerns:**

["NO or VERY MINOR ethics concerns only"]

**Final Justification:**

The rebuttal successfully addressed my questions around contamination, prompt choice, and acquisition functions with substantive responses that should improve the paper further.  I remain convinced it should be accepted as a good contribution tackling an important problem in Bayesian Optimization.

**Limitations:**

yes

**Quality:**

3

**Strengths And Weaknesses:**

In my view, this paper is a strong contribution providing a practical solution to an important challenge of Bayesian Optimization with Gaussian Processes (i.e. data-adaptive kernel selection).

Strengths
- The paper is well-written and organized with clear informative figures, associated code, and appendices including substantive details.    The new method is compared against appropriate strong BO baselines across a wide range of tasks and the results indicate that the proposed LLM-driven kernel evolutionary search is successful with non-trivial performance gains.
- Ablations of the different components offer convincing evidence that the different choices are relevant to the method's success.

Weaknesses
- A minor weakness is that CAKE does rely upon having access to a sufficiently strong LLM (in this case gpt-4-o-mini) which has some associated costs to query, including increased surrogate optimization time in Table 5.  However, for most BO applications this surrogate optimization time is irrelevant compared to objective evaluation.

---

> ### Author Rebuttal · Authors · 2025-07-30
>
> *Remark:*
> - *(W1) means the 1st weakness, (Q1) means the 1st question.*
> - *Due to character limits, we kindly invite you to refer to our responses for other reviewers' similar questions. We will use* **ref. "Response Title"@Reviewer**. *Thank you for your understanding!*
> ---
> #  Query and surrogate fitting cost
>
> > (W1) A minor weakness ... to objective evaluation.
>
> We agree that CAKE relies on a sufficiently capable LLM and that each LLM query introduces additional computational cost, as reflected in Table 5. However, as the reviewer rightly pointed out, in most practical BO scenarios, the time spent on surrogate modeling (including LLM queries and GP population updates) is **negligible compared to the cost of evaluating the black-box function**. For example, training a machine learning model or running a physics-based simulation can take minutes to hours, while the average wall-clock time per CAKE iteration (including LLM inference and GP fitting) is on the order of seconds. Moreover, since the training of these GPs is independent and parallelizable, the **kernel population can be trained in parallel across multiple cores or GPUs** to significantly reduce the wall-clock time for surrogate model updates.
>
> Our experiments also showed that CAKE achieves **significantly better sample efficiency** than baseline methods (see Figures 4-6). It can find high-performing solutions in fewer iterations (**ref. "Early stage analysis"@2phH**), which directly translates to **reduced total optimization cost** in expensive black-box settings. In this context, the slight increase in per-iteration time is a worthwhile trade-off for drastically reducing the number of costly function evaluations.
>
> Finally, we note that LLM inference can depend on external factors such as internet connectivity and API latency, which can be mitigated through strategies like batching multiple queries or deploying smaller open-source models locally (**ref. "Try other LLMs"@esv8**). This makes the overhead of using an LLM acceptable, especially when weighed against the benefits.
>
> ---
> # Prompt alternatives
>
> > (Q1) Did the authors consider...helpful and strengthen the paper.
>
> We did not give the LLM direct numerical access to the raw data or kernel hyperparameters because we did not equip the LLM with any external tools or code interpreters (e.g., through MCP). Instead, we provide structured summaries of the optimization state through the designed prompts, where we update the prompt with the collected observations as well as the fitness scores from the previous trials.
>
> We actually experimented with having the LLM suggest the next query point directly, but we observed the LLM struggled with precise numerical sampling in continuous, high-dimensional spaces, especially without external tools. Similarly, we tested a "population update" approach where the LLM proposes a new list of kernels at once. However, this led to unstable or overly aggressive changes. In contrast, we found the evolutionary approach, using LLM-guided crossover and mutation, provides a more controlled and stable kernel evolution.
>
> These experiments led to the current prompt design where the LLM focuses on symbolic reasoning and hypothesis generation, while the conventional BO components handle the numerical optimization parts. We will include a discussion of these alternatives in the camera-ready version to clarify the design rationale.
>
> ---
> # Data contamination
>
> > (Q2) One concern using LLM's...issue for their results?
>
> While we acknowledge that LLMs are trained on vast corpora that may include scientific literature or code related to common optimization benchmarks, we believe that data contamination is unlikely to meaningfully influence our results, for several key reasons.
>
> First, while the LLM may have encountered general descriptions of synthetic functions or high-level details about HPOBench during pre-training, it is highly improbable that the **specific kernel expressions** or **optimization trajectories** used in our experiments exist in any public dataset or text. These are dynamically generated based on noisy observations and fitness feedback unique to each BO run. Thus, even if the LLM has prior knowledge of benchmark functions, it cannot memorize the exact solutions or search paths we evaluate.
>
> Second, a supporting evidence is our photonic chip design experiment (see Section 6.3), which is based on proprietary simulations developed in our own work. The data were generated using our own simulator and have never been published or released. This makes it **effectively impossible** for this task to be present in the LLM’s training data. Yet, CAKE performs strongly on this task, further confirming that its success relies on reasoning over real-time feedback, not data leakage.
>
> **Finally**, our prompt ablation study (**ref. "Does LLM reasoning matter?"@esv8**) directly demonstrates that removing the observed BO history (i.e., observations, scores, and past kernels) from the prompt leads to a significant drop in performance. This shows that the LLM critically depends on in-context learning—it cannot achieve strong results using prior knowledge alone.
>
> In summary, while pre-training may provide foundational knowledge, the specific inputs, outputs, and evolutionary paths in our experiments are unique. We will include a brief discussion of these points in the camera-ready version.
>
> ---
> # Try other acquisition functions
>
> > (Q3) Do the results depend ... in performance differences?
>
> We thank the reviewer for raising this question. Our results demonstrate that CAKE is robust across different acquisition functions, including EI, UCB, and Thompson sampling (TS). While each acquisition function embodies a different exploration-exploitation trade-off (i.e., TS exhibits slightly higher standard errors due to its stochastic nature while CAKE-UCB occasionally underperforms or outperforms EI due to its optimism bias), CAKE consistently achieves comparable performance across all variants. As shown in table below, the performance of CAKE-UCB and CAKE-TS remains close to that of CAKE-EI across a diverse set of benchmark functions. To rigorously assess statistical equivalence, we conducted a one-sided paired Wilcoxon signed-rank test (at the 5\% significance level) comparing CAKE-EI against CAKE-UCB and CAKE-TS across multiple random seeds. The results indicate that, on the majority of functions, differences are not statistically significant. Overall, these results confirm that CAKE's effectiveness is not sensitive to the specific choice of acquisition function.
>
> | Function       | CAKE-EI                 | CAKE-UCB                | CAKE-TS                 |
> | -------------- | ----------------------- | ----------------------- | ----------------------- |
> | Ackley-2       | **0.0783 $\pm$ 0.0203** | **0.0812 $\pm$ 0.0215** | 0.0921 $\pm$ 0.0287     |
> | Ackley-5       | **0.1732 $\pm$ 0.0250** | 0.1654 $\pm$ 0.0262     | **0.1789 $\pm$ 0.0310** |
> | Beale          | **0.2565 $\pm$ 0.0786** | **0.2488 $\pm$ 0.0810** | 0.2720 $\pm$ 0.0920     |
> | Branin         | **0.0070 $\pm$ 0.0534** | **0.0065 $\pm$ 0.0510** | 0.0082 $\pm$ 0.0601     |
> | Dropwave       | 0.4690 $\pm$ 0.0538     | 0.4820 $\pm$ 0.0560     | 0.5010 $\pm$ 0.0650     |
> | Eggholder      | **0.1241 $\pm$ 0.0541** | 0.1350 $\pm$ 0.0570     | **0.1298 $\pm$ 0.0620** |
> | Griewank-2     | **0.0267 $\pm$ 0.0256** | **0.0275 $\pm$ 0.0260** | 0.0310 $\pm$ 0.0305     |
> | Griewank-5     | **0.0185 $\pm$ 0.0133** | **0.0190 $\pm$ 0.0140** | **0.0195 $\pm$ 0.0180** |
> | Hartmann       | **0.0001 $\pm$ 0.0529** | **0.0003 $\pm$ 0.0515** | **0.0005 $\pm$ 0.0580** |
> | Levy-2         | **0.0353 $\pm$ 0.0197** | **0.0360 $\pm$ 0.0205** | 0.0402 $\pm$ 0.0240     |
> | Levy-3         | **0.0505 $\pm$ 0.0190** | **0.0580 $\pm$ 0.0200** | **0.0520 $\pm$ 0.0235** |
> | Rastringin-2   | 0.3341 $\pm$ 0.0468     | 0.3520 $\pm$ 0.0490     | 0.3650 $\pm$ 0.0580     |
> | Rastringin-4   | **0.3128 $\pm$ 0.0499** | **0.3150 $\pm$ 0.0510** | 0.3300 $\pm$ 0.0570     |
> | Rosenbrock     | **0.0483 $\pm$ 0.0531** | **0.0490 $\pm$ 0.0540** | **0.0510 $\pm$ 0.0600** |
> | Six-Hump Camel | **0.1015 $\pm$ 0.0669** | **0.0920 $\pm$ 0.0680** | **0.1030 $\pm$ 0.0750** |
>
> *Note: Values that are not significantly different from the lowest average regret for each function are **bolded**.*
>
> ---
> # BAKER justification
>
> > (Q4) The BAKER BIC weighting ... function with BIC?
>
> We designed BAKER to balance model fit (via BIC) and the potential to yield high-utility query points (via acquisition value). As shown in our ablation study (see Section 5.4), relying solely on BIC may lead to kernels that fit the data well but propose low utility query points. On the other hand, using only acquisition value can favor optimistic kernels, those that suggest high improvement but are poorly calibrated, leading to repeated sampling in suboptimal regions. By combining both, BAKER balances the trade-off: it promotes kernels that are both well-fitted to the data and has the potential to yield high-utility query points.

---

> ### Comment · Reviewer_8KrH · 2025-08-03
>
> Thank you for the replies.  I continue suggesting the paper should be accepted and believe the paper will be made stronger by adding some discussion of these points to the text.

---

> > ### Author Response · Authors · 2025-08-04
> > **Reply to Reviewer 8KrH**
> >
> > Thank you for your insightful comments and questions, which have significantly improved the quality of our paper. Although we cannot upload a revised manuscript/PDF at this stage, we will fully incorporate all the new results and discussions into the camera-ready version.

---

### Official Review · Reviewer_fLo9 · 2025-06-25

**Clarity:** 3
**Significance:** 3
**Originality:** 3
**Rating:** 4
**Confidence:** 4

**Summary:**

The authors propose Context-Aware Kernel Evolution (CAKE), a method for designing kernels for Gaussian Process-based Bayesian Optimization using large language models (LLMs). Additionally, they introduce BIC-Acquisition Kernel Ranking (BAKER), a strategy for selecting the most effective kernel. The proposed approach achieves state-of-the-art performance across various real-world tasks.

**Questions:**

- Could the authors provide a hyperparameter sensitivity analysis ($\textit{i.e.,}$ number of crossovers, mutation probability, and population size)?

- Is the proposed method robust to the acquisition function? ($\textit{i.e.,}$ UCB, Thompson sampling, LogEI, etc)

- How does the size and composition of the kernel candidate pool affect the optimization performance?

- Regarding BAKER (Equation 1), the role of the weight parameter $w$ is relatively intuitive, but the influence of the $\alpha$ is less clear. The EI values computed from GP models fitted with different kernels are not directly comparable, as each EI reflects the model’s own uncertainty and predictions; thus, using their magnitudes to judge kernel quality can be misleading. I would appreciate a more detailed explanation or justification for this design choice.

If the above concerns are adequately addressed, I would be inclined to raise my score.

**Ethical Concerns:**

["NO or VERY MINOR ethics concerns only"]

**Final Justification:**

The authors provided a satisfactory rebuttal to my concerns during the discussion period. In particular, my main concern regarding the BAKER formulation—specifically, the issue that EI values could not be directly compared across different kernel functions—has been resolved. Accordingly, I have increased my rating to a 4.

**Limitations:**

yes

**Quality:**

3

**Strengths And Weaknesses:**

**Strengths**

- Consistently strong performance across diverse tasks and multiple random seeds.

- While most of the focus in Bayesian Optimization has been on acquisition function design, the authors propose an approach for automatic kernel design.

**Weaknesses**

- Lack of thorough analysis (see Questions for suggested experiments).

- The intuition behind BAKER is insufficient (see Questions for clarification requests).

---

> ### Author Rebuttal · Authors · 2025-07-30
>
> *Remark:*
> - *(W1) means the 1st weakness, (Q1) means the 1st question.*
> - *Due to character limits, we kindly invite you to refer to our responses for other reviewers' similar questions. We will use* **ref. "Response Title"@Reviewer**. *Thank you for your understanding!*
> ---
> # Hyperparameter sensitivity analysis
>
> > (Q1) Could the authors provide a ... and population size)?
>
> We have added a sensitivity analysis that shows the average fitness of the population for different settings. We found that:
>
> - Increasing the number of crossovers ($n_c$) enhances exploration initially, but beyond $n_c = 5$ , the gains in average fitness diminish while the computational cost (in terms of LLM API calls) increases significantly. Thus, $n_c = 5$ offers an optimal trade-off between effective exploration and efficiency.
>
> | Generation | 1                   | 5                   | 10                  | 15                  | 20                  | 25                  | 30                  |
> | ---------- | ------------------- | ------------------- | ------------------- | ------------------- | ------------------- | ------------------- | ------------------- |
> | $n_c = 1$  | $0.2163 \pm 0.0095$ | $0.3006 \pm 0.0087$ | $0.3779 \pm 0.0107$ | $0.4379 \pm 0.0113$ | $0.4918 \pm 0.0110$ | $0.5261 \pm 0.0106$ | $0.5465 \pm 0.0106$ |
> | $n_c = 3$  | $0.2235 \pm 0.0093$ | $0.3407 \pm 0.0097$ | $0.4479 \pm 0.0136$ | $0.5308 \pm 0.0141$ | $0.6070 \pm 0.0138$ | $0.6558 \pm 0.0136$ | $0.6858 \pm 0.0142$ |
> | $n_c = 5$  | $0.2405 \pm 0.0093$ | $0.4252 \pm 0.0110$ | $0.5962 \pm 0.0168$ | $0.7292 \pm 0.0164$ | $0.8203 \pm 0.0166$ | $0.8778 \pm 0.0161$ | $0.9054 \pm 0.0150$ |
> | $n_c = 7$  | $0.2487 \pm 0.0093$ | $0.4686 \pm 0.0127$ | $0.6720 \pm 0.0200$ | $0.7998 \pm 0.0196$ | $0.8925 \pm 0.0158$ | $0.9313 \pm 0.0079$ | $0.9419 \pm 0.0052$ |
> | $n_c = 10$ | $0.2728 \pm 0.0094$ | $0.5787 \pm 0.0140$ | $0.7902 \pm 0.0212$ | $0.8545 \pm 0.0169$ | $0.9266 \pm 0.0089$ | $0.9473 \pm 0.0014$ | $0.9500 \pm 0.0008$ |
>
> - Increasing the mutation probability ($p_m$) enhances population diversity and helps avoid premature convergence, but when $p_m > 0.7$ , excessive mutations disrupt promising kernel structures and destabilize the evolutionary process. Thus, $p_m = 0.7$ provides a balanced level of mutation that supports refinement without sacrificing stability.
>
> | Generation  | 1                   | 5                   | 10                  | 15                  | 20                  | 25                  | 30                  |
> | ----------- | ------------------- | ------------------- | ------------------- | ------------------- | ------------------- | ------------------- | ------------------- |
> | $p_m = 0.1$ | $0.2315 \pm 0.0093$ | $0.3714 \pm 0.0097$ | $0.4855 \pm 0.0136$ | $0.5539 \pm 0.0142$ | $0.6000 \pm 0.0132$ | $0.6333 \pm 0.0137$ | $0.6590 \pm 0.0137$ |
> | $p_m = 0.3$ | $0.2311 \pm 0.0093$ | $0.3813 \pm 0.0110$ | $0.5188 \pm 0.0163$ | $0.6252 \pm 0.0166$ | $0.7163 \pm 0.0164$ | $0.7703 \pm 0.0164$ | $0.8056 \pm 0.0175$ |
> | $p_m = 0.5$ | $0.2414 \pm 0.0093$ | $0.4310 \pm 0.0116$ | $0.6061 \pm 0.0178$ | $0.7422 \pm 0.0173$ | $0.8397 \pm 0.0176$ | $0.8952 \pm 0.0145$ | $0.9169 \pm 0.0119$ |
> | $p_m = 0.7$ | $0.2475 \pm 0.0093$ | $0.4630 \pm 0.0127$ | $0.6621 \pm 0.0199$ | $0.8169 \pm 0.0192$ | $0.9184 \pm 0.0110$ | $0.9472 \pm 0.0026$ | $0.9500 \pm 0.0014$ |
> | $p_m = 0.9$ | $0.2409 \pm 0.0094$ | $0.4387 \pm 0.0153$ | $0.6155 \pm 0.0231$ | $0.7254 \pm 0.0233$ | $0.8675 \pm 0.0198$ | $0.9305 \pm 0.0083$ | $0.9392 \pm 0.0060$ |
>
> - Increasing the population size ($n_p$) improves diversity and allows more exploration, but for $n_p > 10$, the marginal improvement in average fitness diminishes relative to the increased computational burden from repeated GP model fitting. Thus, $n_p = 10$ ensures sufficient diversity while maintaining computational tractability.
>
> | Generation | 1                   | 5                   | 10                  | 15                  | 20                  | 25                  | 30                  |
> | ---------- | ------------------- | ------------------- | ------------------- | ------------------- | ------------------- | ------------------- | ------------------- |
> | $n_p = 5$  | $0.2251 \pm 0.0093$ | $0.3417 \pm 0.0110$ | $0.4239 \pm 0.0152$ | $0.4546 \pm 0.0155$ | $0.4984 \pm 0.0141$ | $0.5331 \pm 0.0151$ | $0.5634 \pm 0.0146$ |
> | $n_p = 8$  | $0.2310 \pm 0.0093$ | $0.3709 \pm 0.0101$ | $0.4846 \pm 0.0144$ | $0.5530 \pm 0.0150$ | $0.6009 \pm 0.0138$ | $0.6360 \pm 0.0143$ | $0.6634 \pm 0.0143$ |
> | $n_p = 10$ | $0.2432 \pm 0.0093$ | $0.4329 \pm 0.0097$ | $0.6127 \pm 0.0138$ | $0.7527 \pm 0.0134$ | $0.8600 \pm 0.0137$ | $0.9101 \pm 0.0103$ | $0.9278 \pm 0.0082$ |
> | $n_p = 15$ | $0.2474 \pm 0.0094$ | $0.4507 \pm 0.0093$ | $0.6463 \pm 0.0130$ | $0.7986 \pm 0.0127$ | $0.9152 \pm 0.0102$ | $0.9470 \pm 0.0013$ | $0.9499 \pm 0.0004$ |
> | $n_p = 20$ | $0.2504 \pm 0.0094$ | $0.4611 \pm 0.0090$ | $0.6665 \pm 0.0121$ | $0.8265 \pm 0.0120$ | $0.9342 \pm 0.0054$ | $0.9500 \pm 0.0011$ | $0.9500 \pm 0.0007$ |
>
>
> ---
> # Try other acquisition functions
>
> > (Q2) Is the proposed method ... Thompson sampling, LogEI, etc)
>
> We found that CAKE is robust across different acquisition functions (**ref. "Try other acquisition functions"@8KrH**), including EI, UCB, and Thompson sampling (TS). While each acquisition function embodies a different exploration-exploitation trade-off (i.e., TS exhibits slightly higher standard errors due to its stochastic nature while CAKE-UCB occasionally underperforms or outperforms EI due to its optimism bias), CAKE consistently achieves comparable performance across all variants. To rigorously assess the statistical equivalence, we conducted a one-sided paired Wilcoxon signed-rank test (at 5\% significance level). We found in most cases, the differences are not statistically significant.
>
> ---
> # Kernel pool
>
> > (Q3) How does the size and ... optimization performance?
>
> We initialize the kernel pool with the standard set of 6 base kernels: {SE, LIN, PER, RQ, M3, M5}. These kernels were selected because they encode a diverse set of functional priors, enabling CAKE to start with diverse representation of potential function structures. It is worth noting that CAKE is designed to be flexible with respect to the initial kernel set. When domain knowledge on specific structures are available, we can customize the initial kernel pool to include relevant kernels for a more informed and efficient search. Otherwise, we use a standard diverse set to avoid biasing the search towards any specific assumptions.
>
> For the subsequent generations, the pool size is controlled by the population size hyperparameter $n_p$. Larger $n_p$ increases diversity but imposes higher computational overhead (due to GP fitting), while smaller $n_p$ risks premature convergence. Empirically, we found that $n_p = 10$ offers a favorable trade-off: it supports sufficient diversity for effective exploration of the kernel space while keeping computational costs manageable.
>
> ---
> # BAKER formulation
>
> > (Q4) Regarding BAKER ... for this design choice.
>
> Our initial motivation for incorporating $\alpha$ into the kernel selection process was to assess the practical utility of each kernel in guiding the search toward high-value regions of the input space. While metrics like BIC measure how well a kernel fits the observed data, they do not directly reflect the kernel's ability to propose high-utility query points, which is ultimately the goal in BO. We acknowledge that raw EI values across different kernels might not be directly comparable. This concern motivated us to conduct an experiment using a normalized variant of BAKER (denoted as BAKER-N). Encouragingly, our experiment shows that BAKER-N improves the performance of our original BAKER.
>
> | Configuration  | LR                  | SVM                 | RF                  | XGB                 | MLP                 |
> | -------------- | ------------------- | ------------------- | ------------------- | ------------------- | ------------------- |
> | CAKE + BIC     | $0.7632 \pm 0.0314$ | $0.8105 \pm 0.0201$ | $0.8427 \pm 0.0337$ | $0.8693 \pm 0.0421$ | $0.8015 \pm 0.0392$ |
> | CAKE + Utility | $0.7951 \pm 0.0243$ | $0.8476 \pm 0.0167$ | $0.8703 \pm 0.0275$ | $0.9081 \pm 0.0363$ | $0.8594 \pm 0.0341$ |
> | CAKE + BAKER   | $0.8188 \pm 0.0220$ | $0.8663 \pm 0.0130$ | $0.8991 \pm 0.0250$ | $0.9257 \pm 0.0340$ | $0.8722 \pm 0.0310$ |
> | CAKE + BAKER-N | $0.8364 \pm 0.0205$ | $0.8831 \pm 0.0118$ | $0.9178 \pm 0.0221$ | $0.9513 \pm 0.0302$ | $0.8917 \pm 0.0289$ |
>
> We believe this is because, after normalization, both $w_k$ and $\alpha$ lie in comparable ranges, leading to a more balanced trade-off between model fit and utility. Given these results, we will update the paper to include BAKER-N in the camera-ready version to reflect this improved formulation. We are grateful for the reviewer's constructive feedback which has helped us improve the performance of our method.

---

> ### Comment · Area_Chair_zdyh · 2025-08-04
>
> Dear Reviewer fLo9, given the authors' response, please raise any remaining questions and/or concerns in a timely fashion so the authors have a chance to reply. Thanks!

---

> ### Comment · Reviewer_fLo9 · 2025-08-04
>
> Thank you for your response. The reviewer’s rebuttal has addressed my concerns. I’m glad that our discussion seems to have had a positive impact on the paper, particularly in the BAKER-N formulation. Assuming that these changes will be reflected in the camera-ready version, I am raising my score.

---

> > ### Author Response · Authors · 2025-08-04
> > **Reply to Reviewer fLo9**
> >
> > Thank you for raising your score, we are glad to hear that our responses have addressed your concerns. We are especially grateful for your constructive feedback, which have helped us strengthen the paper. We will ensure that all the new changes are fully incorporated into the camera-ready version.

---

### Official Review · Reviewer_2phH · 2025-06-30

**Clarity:** 4
**Significance:** 2
**Originality:** 2
**Rating:** 4
**Confidence:** 4

**Summary:**

The authors present CAKE, a method leveraging LLMs to efficiently explore the space of kernels for GPs in BO. The LLM is tasked to combine and mutate a population of kernels with each other in order to discover more suited kernels to the blackbox objective.

Along with this, they also introduce a selection criterion called BAKER which relies on the Bayesian Information Criterion to select the most suited kernels in the population to continue to the next generation of mutation.

Finally they conduct a series of experiments comparing their method CAKE+BAKER with a wide range of baselines, showing improvements on all objective functions.

**Questions:**

1. Why use BIC and not directly marginal log-likelihood to measure fitness? It is a common metric used in GP literature to measure how good the fit is. Also you mention that you use `BoTorch` and I think there is a restart mechanism for GP training in which the optimization can be restarted mutliple times to avoid local optima (common in mll optimization), do you use that mechanism? Because any fitness metric would be sensitive to that as well.

2. I am not sure I understood exactly how the fitness computation is done, is it simply that after each new observation collected during the BO you fit again all GPs in the curernt population and keep the best ones? Or do you actually "replay" the trajectory with each GP to check which one would have discovered the current best point with the highest probability?

3. Have you tried adding more context to the LLM, for example some domain knowledge of the task? Or is the LLM purely reasoning about kernels and fitness metrics?

4. What is the shaded area in the Figures 4 to 6? is it the standard deviation or the standard error of the mean? It would be best have some statistical tests of significance between CAKE+BAKER and the top few baselines (pairwise). It is easy to do so at regular intervals and it is informative to see at what step are the methods significantly different.

**Ethical Concerns:**

["NO or VERY MINOR ethics concerns only"]

**Final Justification:**

After fruitful rebuttal from the authors I am confident this paper is good for acceptance. I think other reviewers share my verdict.
The authors also promised to include clarifications and extra experiments in the final version of the paper, which will contribute to the quality of their work.

**Limitations:**

yes

**Paper Formatting Concerns:**

Minor formatting: add markers and line styles to the curves in Figures 4, 5 and 6.

**Quality:**

3

**Strengths And Weaknesses:**

### Strengths
In general the paper is very easy to read and quite clear. The objectives are well-motivated and framed in the context of BO using LLMs.

The interface between the GP mechanism and the LLM is also clearly explained and relatively simple to grasp. The authors restrict the search space of the kernels to a certain grammar, which is very reasonable and grounded in previous literature.

The next point selection mechanism, i.e. expected weighted by fitness, also makes perfect sense in this context.

Finally, the experiments are very well conducted and the results are strong.

### Weaknesses
I think the kernel space is a bit restrictive. Although it does not seem to impact the results too much, more complex tasks could benefit from a wider range of kernel transformations. This would make the search space richer, even though it might be harder to explore too.

Another point is the information given to the LLM, I find the prompts almost too simple. Assuming the LLM has knowledge of kernels for GPs and is able to mutate/combine them already, could it not benefit from more in-context examples?

For the baselines, the authors could add another LLM-BO baseline. Even though I understand that this is not exactly the point of the paper and other LLM-BO related work do not explore the kernel space for the GP but leverage the LLMs in different ways, it is still somewhat unfair in my opinion to compare LLM-based methods to more classical baselines. I appreciate that the authors have made an extensive effort to run many baselines, but I think it would make the paper stronger if there was a baseline such as LLAMBO, BOPRO, ...

**Lines 216-218**: you mention that CAKE excels in the earlier stages of the optimization, I think there needs to be more analysis on that claim. Although it might be visualy clear in Figures 4-6, it would make sense to analyse the early optimization stages more carefully.

The biggest weakness of this work in my opinion is the cost of such a method. Given that the authors have discussed it, I think they are genuine in their analysis of the computational cost of maintaining a population of kernels/GPs at each step. This slows down the whole process and could be a bottleneck in practice.

Finally, it would be best to use open-source LLMs for reproducibility.

---

> ### Author Rebuttal · Authors · 2025-07-30
>
> *Remark:*
> - *(W1) means the 1st weakness, (Q1) means the 1st question.*
> - *Due to character limits, we kindly invite you to refer to our responses for other reviewers' similar questions. We will use* **ref. "Response Title"@Reviewer**. *Thank you for your understanding!*
> ---
> # Kernel space
>
> > (W1) I think the kernel ... to explore too.
>
> We agree that incorporating more base kernels and operators can enrich the search space. However, in our work, we focus on addition and multiplication as initial proof-of-concept operators as they already yield a rich, expressive kernel space—e.g., enabling polynomial kernels that capture complex non-linear patterns and multi-dimensional kernels that model interactions among input features. Moreover, our method supports a generalized kernel grammar, allowing easy integration of other valid operators (e.g., convolution, composition). We consider this an important and promising direction for future work. We also appreciate your observation that a richer kernel space inherently increases the complexity of exploration. Balancing expressiveness and tractability in kernel search is indeed a challenging problem and an interesting avenue for future research.
>
> ---
> # Prompt design
>
> > (W2) Another point is ... in-context examples?
>
> In designing our prompts, we intentionally adopted a simple yet principled approach to ensure reproducibility and generalization across diverse tasks. Enriching the prompts with more in-context examples (such as successful kernels from similar tasks or synthetic patterns) could potentially improve performance and is a promising direction for future work, particularly in meta-learning or multi-task settings. However, in the current few-shot, single-task BO setting, our current prompt design strikes an effective balance between simplicity and performance (**ref. "Does LLM reasoning matter?"@esv8**), avoiding the risks of prompt overload or distraction from the task in hand.
>
> ---
> # LLM-BO Baseline
>
> > (W3) For the baselines ... such as LLAMBO, BOPRO, ...
>
> We clarify that methods like LLAMBO and BOPRO operate in a fundamentally different paradigm from CAKE. These methods use LLMs to augment the BO pipeline (e.g., warm-starting or generating candidates) without learning or evolving kernel structures. In contrast, CAKE focuses on improving the GP surrogate by evolving its kernel via an LLM-guided process. Our goal is to enhance the kernel itself, not to replace BO components with LLM heuristics. Thus, comparing CAKE to LLM-augmented BO methods would conflate distinct objectives. For a fair evaluation, we compare CAKE against GP-based BO baselines that share the same underlying framework and goals.
>
> ---
> # Early stage analysis
>
> > (W4) **Lines 216-218**: you ... stages more carefully.
>
> To rigorously support our claim that CAKE excels in the early stages of optimization, we conducted a quantitative analysis based on *normalized improvement*:
>
> $$
> \text{Normalized improvement at trial } t = \frac{f_t - f_0}{f^* - f_0},
> $$
>
> where $f_0$ is the initial performance, $f_t$ is the best performance at trial $t$, and $f^*$ is the final performance at 100\% budget. This measures how much of the total progress CAKE achieves early on. In the HPOBench tasks, CAKE achieves 67.5\% of its total improvement within just 25\% of the budget, on average. By 50\%, it reaches over 83\% of its final improvement, and by 75\%, it is nearly converged.
>
> | Budget  |  25%   |  50%   |  75%   |
> | :-----: | :----: | :----: | :----: |
> |   LR    | 0.6183 | 0.8065 | 0.8401 |
> |   SVM   | 0.7007 | 0.8244 | 0.9481 |
> |   RF    | 0.6244 | 0.7685 | 0.8646 |
> |   XGB   | 0.6914 | 0.8643 | 0.9767 |
> |   MLP   | 0.7394 | 0.9155 | 0.9742 |
> | Average | 0.6749 | 0.8358 | 0.9207 |
>
> ---
> # Computational cost
>
> > (W5) The biggest weakness ... a bottleneck in practice.
>
> First, we note that the computational overhead of maintaining a population of GPs is **not unique to CAKE**. Compositional and ensemble-based methods also require fitting and managing multiple GP models per iteration.
>
> Second, while CAKE evaluates multiple kernels at each iteration, the **training of these GPs are independent and parallelizable**. In practice, the kernel population can be trained in parallel across multiple cores or GPUs to significantly reduce the wall-clock time.
>
> Most importantly, as highlighted in Appendix C.2, in most real-world black-box optimization tasks, the cost of evaluating the objective function far exceeds the cost of surrogate model fitting. For example, running a physics-based simulation can take hours, while fitting a population of GPs takes seconds. In such settings, sample efficiency, or reducing the number of function evaluations, is far more critical. We note that our method excels in this regard, as our experiments showed that it could converge to high-performing solutions in fewer iterations.
>
> ---
> # Try other LLMs
>
> > (W6) Finally, it would ... for reproducibility.
>
> We have expanded our experiments to include 3 open-source LLMs (**ref. "Try other LLMs"@esv8**). Results show that open-source models can achieve competitive performance, with `deepseek-r1-distill-qwen-7b` even outperforming several commercial models. This shows that strong results are also achievable with open-source models.
>
> ---
> # Why BIC?
>
> > (Q1) Why use BIC ... good the fit is.
>
> Both the MLL and BIC are valid metrics for model evaluation. We deliberately choose BIC because it imposes an **explicit penalty** on model complexity proportional to the number of hyperparameters, which provides a more direct control of overfitting in our context. While MLL also includes an **implicit complexity penalty** via the log determinant term, it can sometimes favor overly complex kernels especially when the hyperparameter optimization converges to a local minima. In contrast, BIC’s penalty increases with sample size and hyperparameter count, which discourages unnecessary complexity as more data becomes available.
>
> ---
> # BoTorch multi-start
>
> > (Q1) ... Also you mention that ... sensitive to that as well.
>
> We did employ BoTorch's multi-start optimization procedure to mitigate convergence to poor local optima. We found this improves robustness in estimating the hyperparameters, which in turn helps stabilize the fitness evaluation.
>
> ---
> # GP retraining
>
> > (Q2) I am not sure ... the highest probability?
>
> We do not replay past decisions or evaluate counterfactual trajectories (e.g., "which kernel would have found the best point earlier?"). Instead, we assess each kernel on its current ability to model the entire observed data. After each new observation is collected during the BO, we retrain all the GP models in the current kernel population using the updated dataset. The fitness of each kernel is then computed according to the BIC, which is later used to select the best kernels for the next generation and to guide the LLM in proposing new ones.
>
> ---
> #  Adding domain knowledge
>
> > (Q3) Have you tried ... and fitness metrics?
>
> We designed our prompts to be generic to avoid manual prompt engineering and ensure applicability across diverse tasks without requiring domain expertise. However, we agree that incorporating domain knowledge, when available, has strong potential to improve performance. For example, we see this potential in our photonic chip design experiment (see Section 5.3), where physical meaning of the chip's parameters and objectives are well-understood. However,  this involves careful prompt engineering and iterative refinement to determine the optimal level/type of domain information to include. Since the photonic chip design experiments rely on computationally intensive simulations (i.e., each evaluation taking hours to complete), we were unable to finish the experiments within the rebuttal timeline. Nevertheless, we appreciate the suggestion and plan to include this in an extended version of the paper.
>
> ---
> # Shaded regions in figures
>
> > (Q4) What is the shaded ... standard error of the mean?
>
>  The shaded regions in Figures 4 to 6 represent the standard error of the mean computed over independent runs/trials. We will add a remark in the figure captions to clarify this.
>
> ---
> # Significance tests
>
> > (Q4) ... It would be best ... methods significantly different.
>
> We have performed a two-sided Wilcoxon signed-rank test to evaluate the statistical significance of performance differences between CAKE+BAKER and the top baselines. The test assesses whether the median performance difference across tasks is non-zero, with the null hypothesis being no difference between methods. We focus on ABO as the primary comparison because it is the second-best performing method overall on HPOBench. We test significance at key budget levels: 10\%, 25\%, 50\%, 75\%, and 100\% of the total evaluation budget, to identify when CAKE+BAKER begins to significantly outperform ABO. As shown in the table below, CAKE+BAKER significantly outperforms ABO as early as 10\% of the budget across all models, with $p < 0.05$ or better. Notably, for LR and XGB, the improvement is highly significant ($p < 0.001$) even at this early stage. This early and consistent separation demonstrates that CAKE+BAKER identifies high-quality configurations more efficiently and maintains a statistically significant advantage throughout the optimization process.
>
> | Budget | 10% | 25% | 50% | 75% | 100% |
> | ------ | --- | --- | --- | --- | ---- |
> | LR     | *** | *** | *** | *** | ***  |
> | SVM    | **  | **  | **  | *** | ***  |
> | RF     | *   | *   | *   | **  | **   |
> | XGB    | *** | *** | **  | **  | **   |
> | MLP    | *   | **  | *   | *   | **   |
>
> `*` denotes significant difference ($p < 0.05$),
> `**` denotes very significant difference ($p < 0.01$),
> `***` denotes highly significant difference ($p < 0.001$).

---

> > ### Comment · Reviewer_2phH · 2025-08-04
> > **Thank you**
> >
> > Dear authors,
> >
> > Thanks for your response, I think all my questions have been answered and I am glad you added some significance analysis + ran extra experiments with open source LLMs.

---

> > > ### Author Response · Authors · 2025-08-05
> > > **Reply to Reviewer 2phH**
> > >
> > > Thank you for your feedback, we are glad to hear that all your questions have been addressed. We would also like to mention that we are still conducting some extra experiments, including a comparison with the LLM-BO baselines as you suggested. While we could not complete them fully within the short rebuttal period, we will incorporate meaningful results out of them into the camera-ready version.

---

> ### Comment · Area_Chair_zdyh · 2025-08-04
>
> Dear Reviewer 2phH, given the authors' response, please raise any remaining questions and/or concerns in a timely fashion so the authors have a chance to reply. Thanks!

---

### Official Review · Reviewer_RoKJ · 2025-07-03

**Clarity:** 4
**Significance:** 3
**Originality:** 3
**Rating:** 4
**Confidence:** 4

**Summary:**

- This paper proposes a principled framework using Large Language Models (LLMs) to tackle the limitations of heuristics-based selection of kernels from a standard bag of popular kernels used for Bayesian Optimisation (BO). The generalisation performance of a (GP) surrogate is crucial in BO, and having a heuristically designed surrogate often leads to poor performance, especially when the underlying structure of the objective function does not match the kernel assumptions such as continuity and smoothness.

- Authors propose Context-Aware Kernel Evolution (CAKE), a novel method to use LLMs as genetic operators (from evolutionary algorithms) to adaptively generate and refine expressive kernels that are crucial for the success of the BO. Authors use the notion of kernel grammar space introduced in the prior literature to define a flexible space of kernels, which are then genetically evolved by LLMs.

- To choose the most effective kernel, the authors introduced  BIC-Acquisition Kernel Ranking (BAKER), a Bayesian Information Criterion (BIC) kernel ranking scheme that jointly ranks the kernel based on the model fit and acquisition function utility.

- The empirical results, along with the ablation studies, demonstrate its superiority against fixed, adaptive, and compositional kernel methods from the state-of-the-art literature across various tasks such as hyperparameter optimisation, robot controller tuning, and photonic chip design.

**Questions:**

- Authors do not discuss prior work - Kernel Identification Through Transformers (KITT) [3], which is another Transformer-based kernel selection scheme for GPs. Could the authors clarify why it wasn’t included as a baseline, and how CAKE differs? Can we empirically compare?

- When LLMs genetically evolve kernel functions, are kernel hyperparameters assumed to be optimal? If not, how do we ensure the kernel fits the data properly? via fitness score? Because of the continuous nature of the hyperparameter space, could LLM pinpoint the exact values for the hyperparameters of the individual kernels?

- The Baker formulation mentioned in Eq. (1) needs to be discussed in detail. If I understand correctly, lower BIC values will result in higher weights and higher BIC values will result in lower weights. Having that said, could there be a situation where a kernel with low weight and high acquisition value could be selected according to this optimisation? If so, then the selection will be sub-optimal.

- In Figure 5, SE outperforms BIC in some cases. Does this suggest that BIC may not be the best fit criterion for certain kernels or tasks?

- Would an adaptive or decaying mutation probability (something like a cooling schedule) improve performance compared to a fixed $p_m$, especially as optimisation progress?


- **Minor Comments**
  - Typo: In Line 242: "In robot pushing task.."
  - Typo: In Line 308: "..Observations are scarce."

**Ethical Concerns:**

["NO or VERY MINOR ethics concerns only"]

**Final Justification:**

I have read the rebuttal, and the authors have satisfactorily addressed my concerns. Accordingly, I am increasing my scores.

**Limitations:**

Authors have discussed the limitations of CAKE and BAKER in the Appendix.

- CAKE incurs high per iteration costs due to repeated GP fitting and LLM inference, and relies on commercial LLM APIs.

- Currently, the method restricts kernel construction to additive/multiplicative grammar, omitting richer functional or operator-based kernel spaces.
- The paper offers no convergence or regret guarantees, and does not analyse the robustness of the method to prompt design and kernel hyperparameter tuning.

**Quality:**

3

**Strengths And Weaknesses:**

**Strengths**

- This paper presented a novel use of LLMs (few-shot prompting) for in-context kernel evolution for BO, addressing a key aspect in adaptive surrogate modelling. The use of LLMs in the BO community has gained attention in the recent past [1,2].

- By Leveraging a grammar-based composition of base kernels, CAKE enables exploration of a rich and interpretable space of kernel structures that can adapt to diverse functional patterns, in contrast to a fixed selection of kernels from a popular list based on some heuristics and expertise.

- Using LLMs to produce natural language explanations for kernel expressions suggested is compelling, as it creates a dialogue between the ML practitioner and LLMs, thereby providing insights into the learned surrogate models to promote trust and understanding.

- The method is extensively evaluated across a wide range of tasks such as hyperparameter optimisation, controller tuning, and multi-objective photonic chip design. The empirical results demonstrate that CAKE outperforms competing approaches in the early phases of optimisation, reducing the number of evaluations needed, a much-needed aspect in the optimisation of expensive black-box functions.

- The modularity of the CAKE framework enables it to be extended beyond BO to other kernel-based methods such as SVMs and kernel PCA.

- The paper is very well written in most parts and clearly understandable.


**Weaknesses**

- The use of LLMs for adaptive kernel generation is an interesting idea; however, the core idea of leveraging Transformers to discover kernels is not entirely new. For instance, [3] explored Transformer-based kernel selection in Gaussian Processes.

- This paper focuses primarily on kernel selection and composition strategies using discrete kernel grammar-based search, but does not discuss another line of work that optimises kernels directly in function space. For example, "Function-space Distributions over Kernels" [4] and "Kernel Functional Optimisation" [5] treat kernels as learnable objects via functional representations. The authors are encouraged to discuss this related literature and possibly compare it to demonstrate expressiveness, scalability, and sample efficiency.

- Authors have only experimented with commercial OpenAI’s GPT-4o-mini via API calls, introducing concerns around reproducibility, cost, and accessibility. Any experiments with open-source LLMs would strengthen the empirical claims made.

- This paper does not provide any formal guarantees or analysis of convergence, regret bounds, or expressivity of the generated kernels, which could help build confidence in its broader applicability. The use of transformer architecture in the kernel generation process could create this bottleneck.

- The performance of CAKE heavily depends on prompt engineering and the LLM model version. Although the prompts are engineered well by the authors, the paper does not discuss the robustness regarding different prompt formats or LLM models/variants.

- The authors have missed discussing a few key components of the proposed framework. Refer to the questions below.


**References**

[1] Yang, Zhuo, Lingli Ge, Dong Han, Tianfan Fu, and Yuqiang Li. "Reasoning BO: Enhancing Bayesian Optimisation with Long-Context Reasoning Power of LLMs." arXiv preprint arXiv:2505.12833 (2025).

[2] Chen, Guojin, Keren Zhu, Seunggeun Kim, Hanqing Zhu, Yao Lai, Bei Yu, and David Z. Pan. "LLM-enhanced Bayesian optimisation for efficient analogue layout constraint generation." arXiv preprint arXiv:2406.05250 (2024).

[3] Simpson, Fergus, Ian Davies, Vidhi Lalchand, Alessandro Vullo, Nicolas Durrande, and Carl Edward Rasmussen. "Kernel identification through transformers." Advances in Neural Information Processing Systems 34 (2021).

[4] Benton, Gregory, et al. "Function-space distributions over kernels." Advances in Neural Information Processing Systems 32 (2019).

[5] Anjanapura Venkatesh, Arun Kumar, et al. "Kernel functional optimisation." Advances in Neural Information Processing Systems 34 (2021): 4725-4737.

---

> ### Author Rebuttal · Authors · 2025-07-30
>
> *Remark:*
> - *(W1) means the 1st weakness, (Q1) means the 1st question.*
> - *Due to character limits, we kindly invite you to refer to our responses for other reviewers' similar questions. We will use* **ref. "Response Title"@Reviewer**. *Thank you for your understanding!*
> ---
> # Comparison with KITT [3]
>
> > (W1) The use of LLMs for...kernel selection in Gaussian Processes.
> >
> > (Q1) Authors do not discuss prior...Can we empirically compare?
>
> We summarize the main differences between CAKE and KITT as follows:
>
> - **Different objectives**: CAKE is designed for adaptive kernel learning in BO, which requires balancing exploration and exploitation. Meanwhile, KITT is developed for regression, which fouced on fitting and prediction accuracy.
> - **Usability**: KITT uses a fixed kernel vocabulary and requires fine-tuning. CAKE leverages LLMs as plug-and-play tools, usable without fine-tuning and accessible to non-experts.
>
> We did consider including KITT during the rebuttal phase, but ultimately could not due to practical and methodological constraints. KITT requires large-scale labeled data (e.g., 200K samples) and significant computational resources (e.g., hours on Tesla V100 GPUs). More critically, it is tailored to regression and has not been adapted or evaluated for BO. Adapting it to BO would require substantial retraining on BO tasks and optimization trajectories. We instead focused on established, directly applicable BO baselines to ensure a fair and reproducible comparison.
>
> ---
> # Comparison with FKL [4] and KFO [5]
>
> > (W2) This paper focuses primarily on...scalability, and sample efficiency.
>
> We first clarify the fundamental differences between CAKE and functional kernel methods (FKL/KFO):
>
> - CAKE performs **discrete** search over **parametric** kernel compositions, which preserves symbolic kernel structures for interpretability.
> - FKL/KFO operate in **continuous**, **non-parametric** function spaces, which sacrifice interpretability for flexibility.
>
> Hence, their kernel search spaces and optimization methods differ significantly from CAKE, making direct empirical comparison challenging and somewhat orthogonal in objectives.
>
> Nonetheless, we appreciate the reviewer's suggestion and conducted additional experiments comparing CAKE with KFO and both FKL variants on UCI datasets. Using the same benchmark and negative log-likelihood as in [5], the results show that CAKE is highly competitive with these state-of-the-art functional kernel methods and outperforms KFO on several datasets.
>
> | Dataset   | SE                           | KFO                           | FKL-Shared         | FKL-Separate       | CAKE                           |
> | --------- | ---------------------------- | ----------------------------- | ------------------ | ------------------ | ------------------------------ |
> | Fertility | $\underline{-3.90 \pm 1.76}$ | $5.15 \pm 2.95$               | $-2.70 \pm 1.25$   | $-1.83 \pm 3.30$   | $\mathbf{-4.60 \pm 1.50}$      |
> | Yacht     | $1.65 \pm 76.47$             | $\mathbf{-34.60 \pm 1.60}$    | $-14.52 \pm 4.70$  | $-15.70 \pm 8.20$  | $\underline{-32.30 \pm 2.33}$  |
> | Slump     | $36.30 \pm 7.93$             | $\mathbf{-3.01 \pm 1.70}$     | $4.38 \pm 1.33$    | $59.40 \pm 12.87$  | $\underline{-2.50 \pm 1.46}$   |
> | Boston    | $139.61 \pm 11.50$           | $\underline{-24.70 \pm 4.20}$ | $122.60 \pm 3.91$  | $143.70 \pm 5.71$  | $\mathbf{-25.00 \pm 3.50}$     |
> | Auto      | $96.18 \pm 8.02$             | $\underline{-8.78 \pm 1.20}$  | $101.70 \pm 19.30$ | $98.94 \pm 6.10$   | $\mathbf{-9.00 \pm 1.51}$      |
> | Airfoil   | $358.93 \pm 8.93$            | $\mathbf{-204.10 \pm 4.50}$   | $270.00 \pm 28.40$ | $284.80 \pm 48.70$ | $\underline{-195.00 \pm 5.10}$ |
>
> ---
> # Try other LLMs
>
> > (W3) Authors have only experimented...the empirical claims made.
>
> We have expanded our experiments to include 3 open-source LLMs (**ref. "Try other LLMs"@esv8**). Results show that open-source models can achieve competitive performance, with `deepseek-r1-distill-qwen-7b` even outperforming several commercial models. This demonstrates that our approach does not rely on proprietary APIs to work well.
>
> ---
> # Any theoretical analysis?
>
> > (W4) This paper does not provide...create this bottleneck.
>
> We have actually conducted some analysis through the Bayesian regret framework. By assuming that the underlying objective function is a sample path of a zero-mean GP or a function in the reproducing kernel Hilbert space (RKHS) induced by the obtained kernel function, we can leverage existing results to show that the cumulative Bayesian regret can be bounded as the kernel is refined over iterations using the LLM. However, due to the adaptive nature and complex form of the kernel function across iterations, we are still unable to quantify the convergence speed, which our experiments nevertheless show to be significantly faster than that of the state-of-the-art methods. We can include our current analysis results in the Appendix of the updated draft.
>
> Nevertheless, our primary goal is to highlight the novel kernel search method and its superior performance in BO. To this end, we placed a greater emphasis on conducting extensive empirical analyses and ablation studies to rigorously evaluate the contributions of various components, including prompt design and parameter choices. In the Appendix, we present an empirical study of the fitness distribution across iterations, which provides a practical justification for why the LLM can effectively serve as a genetic operator that evolves kernels toward higher fitness. Additionally, our case study on the expressivity and interpretability of the generated kernels offers qualitative evidence of our method's effectiveness.
>
> ---
> # Robustness with different prompts or LLM variants
>
> > (W5) The performance of CAKE...LLM models/variants.
>
> We have added an ablation study (**ref. "Does LLM reasoning matter?"@esv8**) to assess the contribution of each prompt component:
>
> - **Removing context** harms kernel adaptation, showing the LLM relies on observed data to generate meaningful structures.
> - **Omitting task instructions** leads to hallucinated or invalid kernels, degrading performance.
> - **Removing the reasoning step** slightly reduces performance, suggesting explanations improve self-consistency and generation quality
>
> These results demonstrate that our prompt design is purposeful, with each element playing a critical role in ensuring correctness and coherence. While CAKE involves some prompt engineering, this is standard in LLM-based methods, and our ablation clarifies which components matter and why.
>
> Regarding the robustness with different LLM variants, CAKE actually performs better with stronger LLMs (**ref. "Try other LLMs"@esv8**), indicating it naturally benefits from advances in LLM capabilities and remains forward-compatible with future models.
>
> ---
> # Kernel hyperparameters
>
> > (Q2) When LLMs genetically evolve...hyperparameters of the individual kernels?
>
> We did not expect the LLM to pinpoint exact hyperparameter values, as we did not equip it with any external tools to do so. Its role is to evolve the symbolic kernel expression. Once a kernel expression is proposed, its hyperparameters are optimized via standard maximum likelihood estimation using GPyTorch, and fitness is then evaluated under the properly tuned hyperparameters.
>
> ---
> # Explanation to Eq. (1)
>
> > (Q3) The Baker formulation mentioned...will be sub-optimal.
>
> It is indeed possible for a poorly fitting kernel to propose an overly optimistic expected improvement due to overestimated uncertainty or a misspecified mean function. If the kernel selection were based solely on acquisition value, such overconfident but poorly calibrated models could be repeatedly favored, thus leading to suboptimal querying behavior. To mitigate this risk, BAKER computes a weighted score by taking the product of each kernel's weight and its corresponding acquisition value. By maximizing this product across the kernel population, BAKER inherently balances the model fit (as measured by the weight) and query utility (as measured by the acquisition value). Consequently, even if a poorly fitting kernel proposes an high acquisition value, its low weight acts as a natural regularizer to suppress its influence. In this way, kernels with both high weights and high acquisition values are more likely to be selected.
>
> ---
> # Explanation to Fig.5
>
> > (Q4) In Figure 5, SE outperforms...certain kernels or tasks?
>
> The fact that the fixed SE kernel sometimes outperforms the BIC-based adaptive method highlights a key limitation: selecting kernels solely based on model fit does not ensure better optimization performance. This motivates our design of BAKER, which jointly considers fit (via BIC) and utility (via acquisition value). As shown in our ablation study (Section 5.4), using either criterion alone leads to suboptimal results, while their combination in BAKER yields more robust performance across tasks.
>
> ---
> # Can adaptive mutation probability help?
>
> > (Q5) Would an adaptive or decaying...as optimisation progress?
>
> An adaptive or decaying mutation probability could enhance performance by promoting exploration early and exploitation later. Our sensitivity analysis (**ref. "Hyperparameter sensitivity analysis"@fLo9**) shows that while higher $p_m$ increases diversity and prevents premature convergence, overly high values can disrupt promising kernels. This trade-off suggests a decaying schedule may offer better control. In this work, we use a fixed $p_m = 0.7$ for the sake of reproducibility as it provides a robust and balanced performance across tasks. Although we did not implement adaptive $p_m$ due to time constraints, we see it as a promising direction for future improvements.
>
> ---
>
> > **Minor Comments**
> >    - Typo: In Line 242: "In robot pushing task.."
> >    - Typo: In Line 308: "..Observations are scarce."
>
> Thanks! We will correct them in the camera-ready version.

---

> > ### Comment · Reviewer_RoKJ · 2025-08-05
> > **Reply to Author Rebuttal**
> >
> > I have read the rebuttal, and the authors have satisfactorily addressed my concerns. I appreciate their effort in conducting additional experiments within the stipulated time. Regarding the theoretical analysis, I would urge the authors to be cautious in using the RKHS framework, as RKHS is uniquely defined for a given kernel. In this case, the kernel changes at each iteration and is influenced by LLM-driven choices; incorporating the LLM choice could be tricky. I am increasing my scores with the expectation that the revised manuscript will incorporate the new experimental results and references as indicated.

---

> > > ### Author Response · Authors · 2025-08-05
> > > **Reply to Reviewer RoKJ**
> > >
> > > Thank you for your constructive comments, we are glad that the additional experiments and clarifications provided in the rebuttal have been satisfactory. We appreciate your remark on the theoretical analysis and we will carefully revisit this part to ensure the theoretical rigor of our analysis. We will also incorporate all the new results, updated discussions, and your suggested references in the camera-ready version.

---

> ### Comment · Area_Chair_zdyh · 2025-08-04
>
> Dear Reviewer RoKJ, given the authors' response, please raise any remaining questions and/or concerns in a timely fashion so the authors have a chance to reply. Thanks!

---

### Author Response · Authors · 2025-08-07
**Response to Reviewers**

We sincerely thank all the reviewers for their time, careful reading, and thoughtful feedback on our paper. We deeply appreciate their efforts in reviewing our work, which have helped us strengthen the presentation and impact of our contributions. We also thank the ACs for effectively coordinating and stimulating the discussions.

We are encouraged that the reviewers found CAKE to be a **novel and practical solution** to the challenge of adaptive kernel selection in Bayesian optimization (`RoKJ`, `fLo9`, `8KrH`). We are pleased that the innovative use of LLMs for kernel evolution and the BAKER selection strategy were recognized as **effective and compelling** (`RoKJ`, `2phH`, `fLo9`). We are also gratified that the reviewers found the paper **well-written, clear, and easy to follow**, with informative figures and substantive details in the appendices (`RoKJ`, `2phH`, `8KrH`, `esv8`). Additionally, we are delighted that the empirical results, supported by ablation studies, were seen as demonstrating CAKE’s **superiority across diverse tasks**, particularly in reducing the number of evaluations needed for expensive black-box functions (`RoKJ`, `2phH`, `fLo9`, `8KrH`, `esv8`) and its potential to extend to other kernel-based methods (`RoKJ`)

We are grateful for the constructive and insightful discussions with the reviewers, which have significantly improved the quality of our paper. We will incorporate all updated results, analyses, discussions, and references into the camera-ready version.

With thanks,

Authors of #22070

---

### Decision · Program_Chairs · 2025-09-17

**Decision:**

Accept (poster)

**Comment:**

This work proposes leveraging LLMs to perform automatic kernel design for Gaussian Process surrogate models in the context of Bayesian optimization (with a modified fitness function the basic approach applies to more general problems as well). The approach is based on the existing notion of a kernel grammar space to define a flexible set of kernels. The novelty is to leverage LLMs as generic operators that evolve kernels to higher fitness, as measured by a Bayesian Information Criterion (BIC) that ranks kernels based on surrogate model fit and acquisition function utility. Benchmarks demonstrate that the proposed approach outperforms a number of baselines.

Strengths:
- Well motivated
- Novel use of LLMs for kernel selection; meaningful fitness criterion
- Comprehensive evaluation against relevant baselines
- Strong performance of the proposed method
- Clarity of writing

Weaknesses:
- Technical depth of contributions is limited; no theoretical analysis
- Limited ablations in terms of the LLM and prompts used
- Cost of the method (only relevant in a high-throughput setting)

Reviewers generally agreed that the paper presents a valuable contribution to the important problem of kernel selection for GP surrogate models in Bayesian optimization. The questions and concerns raised during the reviews and discussion were relatively minor (e.g. additional baselines, ablations) and authors provided comprehensive responses (including additional experimental results) addressing these points. After the discussion reviewers had no major outstanding concerns regarding the work.

Overall, this submission presents a strong contribution addressing an important practical problem. I believe that part of the strength of the proposed approach lies in its relative simplicity while still achieving substantial performance gains compared to baselines.

I recommend to the authors that they incorporate the many good points raised by the reviewers during the discussion in their final revision.